# MARKOV CHAIN SCORE ASCENT:
# A Unifying Framework of
# Variational Inference with Markovian Gradients

**Kyurae Kim**[*]
Sogang University
msca8h@sogang.ac.kr

**Jisu Oh**[†]
Sogang University
jisuoh@sogang.ac.kr

**Jacob R. Gardner**
University of Pennsylvania
jacobrg@seas.upenn.edu

**Adji Bousso Dieng**
Princeton University
adji@princeton.edu

**Hongseok Kim**[‡]
Sogang University
hongseok@sogang.ac.kr

## Abstract

Minimizing the inclusive Kullback-Leibler (KL) divergence with stochastic gradient descent (SGD) is challenging since its gradient is defined as an integral over the posterior. Recently, multiple methods have been proposed to run SGD with *biased* gradient estimates obtained from a Markov chain. This paper provides the first non-asymptotic convergence analysis of these methods by establishing their mixing rate and gradient variance. To do this, we demonstrate that these methods–which we collectively refer to as Markov chain score ascent (MCSA) methods–can be cast as special cases of the Markov chain gradient descent framework. Furthermore, by leveraging this new understanding, we develop a novel MCSA scheme, *parallel* MCSA (pMCSA), that achieves a tighter bound on the gradient variance. We demonstrate that this improved theoretical result translates to superior empirical performance.

## 1 Introduction

Bayesian inference aims to analyze the posterior distribution of an unknown latent variable $\mathbf{z}$ from which data $\mathbf{x}$ is observed. By assuming a model $p(\mathbf{x}|\mathbf{z})$, the posterior $\pi(\mathbf{z})$ is given by Bayes' rule such that $\pi(\mathbf{z}) \propto p(\mathbf{x}|\mathbf{z}) p(\mathbf{z})$ where $p(\mathbf{z})$ represents our prior belief on $\mathbf{z}$. Instead of working directly with $\pi$, variational inference (VI, Blei *et al.* 2017) seeks a *variational approximation* $q(\mathbf{z};\boldsymbol{\lambda}) \in \mathcal{Q}$, where $\mathcal{Q}$ is a variational family and $\boldsymbol{\lambda}$ are the variational parameters, that is the most similar to $\pi$ according to a discrepancy measure $d(\pi, q(\cdot;\boldsymbol{\lambda}))$.

The apparent importance of choosing the right discrepancy measure has led to a quest spanning a decade (Bamler *et al.*, 2017; Dieng *et al.*, 2017; Geffner & Domke, 2021b; Hernandez-Lobato *et al.*, 2016; Li & Turner, 2016; Regli & Silva, 2018; Ruiz & Titsias, 2019; Salimans *et al.*, 2015; Wan *et al.*, 2020; Wang *et al.*, 2018; Zhang *et al.*, 2021). So far, the exclusive (or reverse, backward) Kullback-Leibler (KL) divergence $d_{\mathrm{KL}}(q(\cdot;\boldsymbol{\lambda}) \parallel \pi)$ has seen "exclusive" use, partly because it is defined as an integral over $q(\mathbf{z};\boldsymbol{\lambda})$, which can be approximated efficiently. In contrast, the *inclusive* (or forward) KL is defined as an integral over $\pi$ as

$$d_{\mathrm{KL}}(\pi \parallel q(\cdot;\boldsymbol{\lambda})) = \int \pi(\mathbf{z}) \log \frac{\pi(\mathbf{z})}{q(\mathbf{z};\boldsymbol{\lambda})} \, d\mathbf{z} = \mathbb{E}_{\mathbf{z} \sim \pi(\cdot)} \left[ \log \frac{\pi(\mathbf{z})}{q(\mathbf{z};\boldsymbol{\lambda})} \right].$$

---

[*]K. Kim is currently with the University of Pennsylvania.

[†]J. Oh is currently with North Carolina State University.

[‡]Corresponding author.

36th Conference on Neural Information Processing Systems (NeurIPS 2022).

Since our goal is to approximate $\pi$ with $q\left(\cdot;\lambda\right)$ but the inclusive KL involves an integral over $\pi$, we end up facing a chicken-and-egg problem. Despite this challenge, the inclusive KL has consistently drawn attention due to its statistical properties, such as better uncertainty estimates due to its mass covering property (MacKay, 2001; Minka, 2005; Trippe & Turner, 2017).

Recently, Naesseth *et al.* (2020); Ou & Song (2020) have respectively proposed Markovian score climbing (MSC) and joint stochastic approximation (JSA). These methods minimize the inclusive KL using stochastic gradient descent (SGD, Robbins & Monro 1951), where the gradients are estimated using a Markov chain. The Markov chain kernel $K_{\lambda_t}\left(\mathbf{z}_t,\cdot\right)$ is $\pi$-invariant (Robert & Casella, 2004) and is chosen such that it directly takes advantage of the current variational approximation $q\left(\cdot;\lambda_t\right)$. Thus, the quality of the gradients improves over time as the KL divergence decreases. Still, the gradients are non-asymptotically biased and Markovian across adjacent iterations, which sharply contrasts MSC and JSA from classical black-box VI (Kucukelbir *et al.*, 2017; Ranganath *et al.*, 2014), where the gradients are unbiased and independent. While Naesseth *et al.* (2020) have shown the convergence of MSC through the work of Gu & Kong (1998), this result is only asymptotic and does not provide practical insight into the performance of MSC.

In this paper, we address these theoretical gaps by casting MSC and JSA into a general framework we call Markov chain score ascent (MCSA), which we show is a special case of Markov chain gradient descent (MCGD, Duchi *et al.* 2012). This enables the application of the non-asymptotic convergence results of MCGD (Debavelaere *et al.*, 2021; Doan *et al.*, 2020a,b; Duchi *et al.*, 2012; Karimi *et al.*, 2019; Sun *et al.*, 2018; Xiong *et al.*, 2021). For MCGD methods, the fundamental properties affecting the convergence rate are the ergodic convergence rate ($\rho$) of the MCMC kernel and the gradient variance ($G$). We analyze $\rho$ and $G$ of MSC and JSA, enabling their practical comparison given a fixed computational budget ($N$). Furthermore, based on the recent insight that the mixing rate does not affect the convergence rate of MCGD (Doan *et al.*, 2020a,b), we propose a novel scheme, parallel MCSA (pMCSA), which achieves lower variance by trading off the mixing rate. We verify our theoretical analysis through numerical simulations and compare MSC, JSA, and pMCSA on general Bayesian inference problems. Our experiments show that our proposed method outperforms previous MCSA approaches.

### Contribution Summary

❶ We provide the first non-asymptotic theoretical analysis of two recently proposed inclusive KL minimization methods (**Section 4**), MSC (**Theorems 1 and 2**) and JSA (**Theorem 3**).

❷ To do this, we show that both methods can be viewed as what we call "Markov chain score ascent" (MCSA) methods (**Section 3**), which are a special case of MCGD (**Proposition 1**).

❸ In light of this, we develop a novel MCSA method which we call parallel MCSA (pMCSA, **Section 5**) that achieves lower gradient variance (**Theorem 4**).

❹ We demonstrate that the improved theoretical performance of pMCSA translates to superior empirical performance across a variety of Bayesian inference tasks (**Section 6**).

## 2 Background

### 2.1 Inclusive Kullback-Leibler Minimization with Stochastic Gradients

**VI with SGD**  The goal of VI is to find the optimal variational parameters $\lambda$ identifying $q\left(\cdot;\lambda\right) \in \mathcal{Q}$ that minimizes some discrepancy measure $D\left(\pi, q\left(\cdot;\lambda\right)\right)$. A typical way to perform VI is to use stochastic gradient descent (SGD, Robbins & Monro 1951), provided that the optimization objective provides *unbiased* gradient estimates $\mathbf{g}\left(\lambda\right)$ such that we can repeat the update

$$\lambda_t = \lambda_{t-1} - \gamma_t\,\mathbf{g}\left(\lambda_{t-1}\right),$$

where $\gamma_1, \ldots, \gamma_T$ is a stepsize schedule.

**Inclusive KL Minimization with SGD**  For inclusive KL minimization, $\mathbf{g}$ should be set as

$$\mathbf{g}\left(\lambda\right) = \nabla_\lambda d_{\mathrm{KL}}\left(\pi \parallel q\left(\cdot;\lambda\right)\right) = \nabla_\lambda \mathbb{H}\left[\pi, q\left(\cdot;\lambda\right)\right] = -\mathbb{E}_{\mathbf{z}\sim\pi(\cdot)}\left[\,\mathbf{s}\left(\lambda;\mathbf{z}\right)\,\right],$$

where $\mathbb{H}\left[\pi, q\left(\cdot;\lambda\right)\right]$ is the cross-entropy between $\pi$ and $q\left(\cdot;\lambda\right)$, which shows the connection with cross-entropy methods (de Boer *et al.*, 2005), and $\mathbf{s}\left(\lambda;\mathbf{z}\right) = \nabla_\lambda \log q\left(\mathbf{z};\lambda\right)$ is known as the *score gradient*. Since inclusive KL minimization with SGD is equivalent to ascending towards the direction of the score, Naesseth *et al.* (2020) coined the term score climbing. To better conform with the optimization literature, we instead call this approach *score ascent* as in gradient ascent.

Table 1: Convergence Rates of MCGD Algorithms

| Algorithm | Stepsize Rule | Gradient Assumption | Rate | Reference |
|---|---|---|---|---|
| Mirror Descent[1] | $\gamma_t = \gamma/\sqrt{t}$ | $\mathbb{E}\left[\,\|\mathbf{g}(\boldsymbol{\lambda},\boldsymbol{\eta})\,\|_*^2 \mid \mathcal{F}_{t-1}\right] < G^2$ | $\mathcal{O}\left(\frac{G^2 \log T}{\log \rho^{-1}\sqrt{T}}\right)$ | Duchi *et al.* (2012) Corollary 3.5 |
| SGD-Nesterov[2] | $\gamma_t = 2/(t+1)$ $\beta_t = \frac{1}{2L\sqrt{t+1}}$ | $\|\mathbf{g}(\boldsymbol{\lambda},\boldsymbol{\eta})\|_2 < G$ | $\mathcal{O}\left(\frac{G^2 \log T}{\sqrt{T}}\right)$ | Doan *et al.* (2020a) Theorem 2 |
| SGD[3] | $\gamma_t = \gamma/t$ $\gamma = \min\{1/2L, 2L/\mu\}$ | $\|\mathbf{g}(\boldsymbol{\lambda},\boldsymbol{\eta})\|_* < G\left(\|\boldsymbol{\lambda}\|_2 + 1\right)$ | $\mathcal{O}\left(\frac{G^2 \log T}{T}\right)$ | Doan *et al.* (2020b) Theorem 1,2 |

**Notation:** [1]$\mathcal{F}_t$ is the $\sigma$-field formed by the iterates $\boldsymbol{\eta}_t$ up to the $t$th MCGD iteration, $\|\mathbf{x}\|_*$ is the dual norm of $\mathbf{x}$; [2]$\beta_t$ is the stepsize of the momentum; [23]$L$ is the Lipschitz smoothness constant; [3]$\mu$ is the strong convexity constant.

## 2.2 Markov Chain Gradient Descent

**Overview of MCGD** Markov chain gradient descent (MCGD, Duchi *et al.* 2012; Sun *et al.* 2018) is a family of algorithms that minimize a function $f$ defined as $f(\boldsymbol{\lambda}) = \int f(\boldsymbol{\lambda},\boldsymbol{\eta})\,\Pi(d\boldsymbol{\eta})$, where $\boldsymbol{\eta}$ is random noise, and $\Pi(d\boldsymbol{\eta})$ is its probability measure. MCGD repeats the steps

$$\boldsymbol{\lambda}_{t+1} = \boldsymbol{\lambda}_t - \gamma_t\,\mathbf{g}(\boldsymbol{\lambda}_t,\boldsymbol{\eta}_t), \quad \boldsymbol{\eta}_t \sim P_{\boldsymbol{\lambda}_{t-1}}(\boldsymbol{\eta}_{t-1},\cdot), \tag{1}$$

where $P_{\boldsymbol{\lambda}_{t-1}}$ is a $\Pi$-invariant Markov chain kernel that may depend on $\boldsymbol{\lambda}_{t-1}$. The noise of the gradient is Markovian and non-asymptotically biased, departing from vanilla SGD. Non-asymptotic convergence of this general algorithm has recently started to gather attention as by Debavelaere *et al.* (2021); Doan *et al.* (2020a,b); Duchi *et al.* (2012); Karimi *et al.* (2019); Sun *et al.* (2018).

**Applications of MCGD** MCGD encompasses an extensive range of problems, including distributed optimization (Ram *et al.*, 2009), reinforcement learning (Doan *et al.*, 2020a; Tadić & Doucet, 2017; Xiong *et al.*, 2021), and expectation-minimization (Karimi *et al.*, 2019), to name a few. This paper extends this list with inclusive KL VI through the MCSA framework.

## 3 Markov Chain Score Ascent

First, we develop Markov chain score ascent (MCSA), a framework for inclusive KL minimization with MCGD. This framework will establish the connection between MSC/JSA and MCGD.

### 3.1 Markov Chain Score Ascent as a Special Case of Markov Chain Gradient Descent

As shown in Equation (1), the basic ingredients of MCGD are the target function $f(\boldsymbol{\lambda},\boldsymbol{\eta})$, the gradient estimator $\mathbf{g}(\boldsymbol{\lambda},\boldsymbol{\eta})$, and the Markov chain kernel $P_{\boldsymbol{\lambda}}(\boldsymbol{\eta},\cdot)$. Obtaining MCSA from MCGD boils down to designing $\mathbf{g}$ and $P_{\boldsymbol{\lambda}}$ such that $f(\boldsymbol{\lambda}) = d_{\mathrm{KL}}(\pi \parallel q(\cdot;\boldsymbol{\lambda}))$. The following proposition provides sufficient conditions on $\mathbf{g}$ and $P_{\boldsymbol{\lambda}}$ to achieve this goal.

**Proposition 1.** Let $\boldsymbol{\eta} = (\mathbf{z}^{(1)}, \mathbf{z}^{(2)}, \ldots, \mathbf{z}^{(N)})$ and a Markov chain kernel $P_{\boldsymbol{\lambda}}(\boldsymbol{\eta},\cdot)$ be $\Pi$-invariant where $\Pi$ is defined as

$$\Pi(\boldsymbol{\eta}) = \pi(\mathbf{z}^{(1)})\,\pi(\mathbf{z}^{(2)}) \times \ldots \times \pi(\mathbf{z}^{(N)}).$$

Then, by defining the objective function $f$ and the gradient estimator $\mathbf{g}$ to be

$$f(\boldsymbol{\lambda},\boldsymbol{\eta}) = -\frac{1}{N}\sum_{n=1}^{N}\log q(\mathbf{z}^{(n)};\boldsymbol{\lambda}) - \mathbb{H}[\pi] \quad \text{and} \quad \mathbf{g}(\boldsymbol{\lambda},\boldsymbol{\eta}) = -\frac{1}{N}\sum_{n=1}^{N}\mathbf{s}(\mathbf{z}^{(n)};\boldsymbol{\lambda}),$$

where $\mathbb{H}[\pi]$ is the entropy of $\pi$, MCGD results in inclusive KL minimization as

$$\mathbb{E}_{\Pi}[f(\boldsymbol{\lambda},\boldsymbol{\eta})] = d_{\mathrm{KL}}(\pi \parallel q(\cdot;\boldsymbol{\lambda})) \quad \text{and} \quad \mathbb{E}_{\Pi}[\mathbf{g}(\boldsymbol{\lambda},\boldsymbol{\eta})] = \nabla_{\boldsymbol{\lambda}}\,d_{\mathrm{KL}}(\pi \parallel q(\cdot;\boldsymbol{\lambda})).$$

*Proof.* See the *full proof* in page 19.

This simple connection between MCGD and VI paves the way toward the non-asymptotic analysis of JSA and MSC. Note that $N$ here can be regarded as the computational budget of each MCGD iteration since the cost of **(i)** generating the Markov chain samples $\mathbf{z}^{(1)}, \ldots, \mathbf{z}^{(N)}$ and **(ii)** computing the gradient $\mathbf{g}$ will linearly increase with $N$.

In addition, the MCGD framework often assumes $P$ to be geometrically ergodic. An exception is the analysis of Debavelaere *et al.* (2021) where they work with polynomially ergodic kernels.

**Assumption 1** (Uniform Geometric Ergodicity). *The Markov chain kernel $P$ is geometrically ergodic as*

$$d_{\mathrm{TV}}\left(P_\lambda^n\left(\eta,\cdot\right),\Pi\right) \le C\,\rho^n$$

*for some positive constant $C$.*

### 3.2 Non-Asymptotic Convergence of Markov Chain Score Ascent

**Non-Asymptotic Convergence** Through Proposition 1, Assumption 1 and some technical assumptions on the objective function, we can apply the existing convergence results of MCGD to MCSA. Table 1 provides a list of relevant results. Apart from properties of the objective function (such as Lipschitz smoothness), the convergence rates are stated in terms of the gradient bound $G$, kernel mixing rate $\rho$, and the number of MCGD iterations $T$. We focus on $G$ and $\rho$ as they are closely related to the design choices of different MCSA algorithms.

**Convergence and the Mixing Rate $\rho$** Duchi *et al.* (2012) was the first to provide an analysis of the general MCGD setting. Their convergence rate is dependent on the mixing rate through the $1/\log\rho^{-1}$ term. For MCSA, this result is overly conservative since, on challenging problems, mixing can be slow such that $\rho \approx 1$. Fortunately, Doan *et al.* (2020a,b) have recently shown that it is possible to obtain a rate independent of the mixing rate $\rho$. For example, in the result of Doan *et al.* (2020b), the influence of $\rho$ decreases in a rate of $\mathcal{O}\left(1/T^2\right)$. This observation is critical since it implies that **trading a "slower mixing rate" for "lower gradient variance" could be profitable**. We exploit this observation in our novel MCSA scheme in Section 5.

**Gradient Bound $G$** Except for Doan *et al.* (2020b), most results assume that the gradient is bounded for $\forall\eta,\lambda$ as $\|\mathbf{g}\left(\lambda,\eta\right)\| < G$. Admittedly, this condition is strong, but it is similar to the bounded variance assumption $\mathbb{E}[\|\mathbf{g}\|^2] < G^2$ used in vanilla SGD, which is also known to be strong as it contradicts strong convexity (Nguyen *et al.*, 2018). Nonetheless, assuming $G$ can have practical benefits beyond theoretical settings. For example, Geffner & Domke (2020) use $G$ to compare the performance different VI gradient estimators. In a similar spirit, we will obtain the gradient bound $G$ of different MCSA algorithms and compare their theoretical performance.

## 4 Demystifying Prior Markov Chain Score Ascent Methods

In this section, we will show that MSC and JSA both qualify as MCSA methods. Furthermore, we establish **(i)** the mixing rate of their implicitly defined kernel $P$ and **(ii)** the upper bound on their gradient variance. This will provide insight into their practical non-asymptotic performance.

### 4.1 Technical Assumptions

To cast previous methods into MCSA, we need some technical assumptions.

**Assumption 2** (Bounded importance weight). *The importance weight ratio $w\left(\mathbf{z}\right) = \pi\left(\mathbf{z}\right)/q\left(\mathbf{z};\lambda\right)$ is bounded by some finite constant as $w^* < \infty$ for all $\lambda \in \Lambda$ such that $r = \left(1 - 1/w^*\right) < 1$.*

This assumption is necessary to ensure Assumption 1, and can be practically ensured by using a variational family with heavy tails (Domke & Sheldon, 2018) or using a defensive mixture (Hesterberg, 1995; Holden *et al.*, 2009) as

$$q_{\mathrm{def.}}\left(\mathbf{z};\lambda\right) = \alpha\,q\left(\mathbf{z};\lambda\right) + \left(1-\alpha\right)\nu\left(\mathbf{z}\right),$$

where $0 < \alpha < 1$ and $\nu\left(\cdot\right)$ is a heavy tailed distribution such that $\sup_{\mathbf{z}\in\mathcal{Z}}\pi\left(\mathbf{z}\right)/\nu\left(\mathbf{z}\right) < \infty$. Note that $q_{\mathrm{def.}}$ is only used in the Markov chain kernels and $q\left(\cdot;\lambda\right)$ is still the output of the VI procedure. While these tricks help escape slowly mixing regions, this benefit quickly vanishes as $\lambda$ converges. Therefore, ensuring Assumption 2 seems unnecessary in practice unless we absolutely care about ergodicity. (Think of the adaptive MCMC setting for example. Brofos *et al.* 2022; Holden *et al.* 2009).

**Model (Variational Family) Misspecification and $w^*$** Note that $w^*$ is bounded below exponentially by the inclusive KL as shown in Proposition 2. Therefore, $w^*$ will be large **(i)** in the initial steps of VI and **(ii)** under model (variational family) misspecification.

**Assumption 3.** *(Bounded Score) The score gradient is bounded for $\forall\lambda \in \Lambda$ and $\forall\mathbf{z} \in \mathcal{Z}$ such that $\|\mathbf{s}\left(\lambda;\mathbf{z}\right)\|_2 \le L$ for some finite constant $L > 0$.*

Although this assumption is strong, it enables us to compare the gradient variance of MCSA methods. We empirically justify the bounds obtained using Assumption 3 in Section 6.2.

## 4.2 Markovian Score Climbing

MSC (Algorithm 4 in Appendix B) is a simple instance of MCSA where $\boldsymbol{\eta}_t = \mathbf{z}_t$ and $P_{\lambda_t} = K_{\lambda_t}$ is the conditional importance sampling (CIS) kernel (originally proposed by Andrieu *et al.* (2018)) where the proposals are generated from $q\left(\cdot; \lambda_t\right)$. Although MSC uses only a single sample for the Markov chain, the CIS kernel internally operates $N - 1$ proposals. Therefore, $N$ in MSC has a different meaning, but it still indicates the computational budget.

**Theorem 1.** MSC (Naesseth *et al.*, 2020) is obtained by defining

$$P_\lambda^k\left(\boldsymbol{\eta}, d\boldsymbol{\eta}'\right) = K_\lambda^k\left(\mathbf{z}, d\mathbf{z}'\right)$$

with $\boldsymbol{\eta}_t = \mathbf{z}_t$, where $K_\lambda\left(\mathbf{z}, \cdot\right)$ is the CIS kernel with $q_{\text{def.}}\left(\cdot; \lambda\right)$ as its proposal distribution. Then, given Assumption 2 and 3, the mixing rate and the gradient bounds are given as

$$d_{\text{TV}}\left(P_\lambda^k\left(\boldsymbol{\eta}, \cdot\right), \Pi\right) \leq \left(1 - \frac{N-1}{2w^*+N-2}\right)^k \quad \text{and} \quad \mathbb{E}\left[\left\|\mathbf{g}_{t,\text{MSC}}\right\|^2 \,\big|\, \mathcal{F}_{t-1}\right] \leq L^2,$$

where $w^* = \sup_{\mathbf{z}} \pi\left(\mathbf{z}\right) / q_{\text{def.}}\left(\mathbf{z}; \lambda\right)$.

*Proof.* See the *full proof* in page 21.

**Discussion**  Theorem 1 shows that the gradient variance of MSC is insensitive to $N$. Although the mixing rate does improve with $N$, when $w^*$ is large due to model misspecification and lack of convergence (see the discussion in Section 4.1), this will be marginal. Overall, *the performance of MSC cannot be improved by increasing the computational budget $N$.*

**Rao-Blackwellization**  Meanwhile, Naesseth *et al.* also provide a Rao-Blackwellized version of MSC we denote as MSC-RB. Instead of selecting a single $\mathbf{z}_t$ by resampling over the $N$ internal proposals, they suggest forming an importance-weighted estimate (Robert & Casella, 2004). The theoretical properties of this estimator have been concurrently analyzed by Cardoso *et al.* (2022).

**Theorem 2.** (Cardoso *et al.*, 2022) The gradient variance of MSC-RB is bounded as

$$\mathbb{E}\left[\left\|\mathbf{g}_{t,\text{MSC-RB}}\right\|_2^2 \,\big|\, \mathcal{F}_{t-1}\right] \leq 4 L^2 \left[\frac{1}{N-1} d_{\chi^2}(\pi \,\|\, q\left(\cdot; \lambda_{t-1}\right)) + \mathcal{O}\left(N^{-3/2} + \gamma^{t-1}/N-1\right)\right] + \|\boldsymbol{\mu}\|_2^2,$$

where $\boldsymbol{\mu} = \mathbb{E}_\pi \mathbf{s}\left(\lambda; \mathbf{z}\right)$, $\gamma = 2w^* / \left(2w^* + N - 2\right)$ is the mixing rate of the Rao-Blackwellized CIS kernel, and $d_{\chi^2}(\pi \,\|\, q) = \int \left(\pi/q - 1\right)^2 q\left(d\mathbf{z}\right)$ is the $\chi^2$ divergence.

*Proof.* See the *full proof* in page 22.

The variance of MSC-RB decreases as $\mathcal{O}\left(1/N-1\right)$, which is more encouraging than vanilla MSC. However, the first term depends on the $\chi^2$ divergence, which is bounded below exponentially by the KL (Agapiou *et al.*, 2017). Therefore, *the variance of MSC-RB will be large on challenging problems where the $\chi^2$ divergence is large*, although linear variance reduction is possible.

## 4.3 Joint Stochastic Approximation

JSA (Algorithm 5 in Appendix B) was proposed for deep generative models where the likelihood factorizes into each datapoint. Then, subsampling can be used through a random-scan version of the independent Metropolis-Hastings (IMH, Hastings 1970) kernel. Instead, we consider the general version of JSA with a vanilla IMH kernel since it can be used for any type of likelihood. At each MCGD step, JSA performs multiple Markov chain transitions and estimates the gradient by averaging all the intermediate states, which is closer to how traditional MCMC is used.

**Independent Metropolis-Hastings**  Similarly to MSC, the IMH kernel in JSA generates proposals from $q\left(\cdot; \lambda_t\right)$. To show the geometric ergodicity of the implicit kernel $P$, we utilize the geometric convergence rate of IMH kernels provided by Mengersen & Tweedie (1996, Theorem 2.1) and Wang (2022). The gradient variance, on the other hand, is difficult to analyze, especially the covariance between the samples. However, we show that, even if we ignore the covariance terms, the variance reduction with respect to $N$ is severely limited in the large $w^*$ regime. To do this, we use the exact $n$-step marginal IMH kernel derived by Smith & Tierney (1996) as

$$K_\lambda^n\left(\mathbf{z}, d\mathbf{z}'\right) = T_n\left(w\left(\mathbf{z}\right) \vee w\left(\mathbf{z}'\right)\right) \pi\left(\mathbf{z}'\right) d\mathbf{z}' + \lambda^n\left(w\left(\mathbf{z}\right)\right) \delta_\mathbf{z}\left(d\mathbf{z}'\right), \tag{2}$$

where $w\left(\mathbf{z}\right) = \pi\left(\mathbf{z}\right) / q_{\text{def.}}\left(\mathbf{z}; \lambda\right)$, $x \vee y = \max\left(x, y\right)$, and for $R\left(v\right) = \left\{\mathbf{z}' \mid w\left(\mathbf{z}'\right) \leq v\right\}$,

$$T_n\left(w\right) = \int_w^\infty \frac{n}{v^2} \lambda^{n-1}\left(v\right) dv \quad \text{and} \quad \lambda\left(w\right) = \int_{R(w)} \left(1 - \frac{w\left(\mathbf{z}'\right)}{w}\right) \pi\left(d\mathbf{z}'\right). \tag{3}$$

**Theorem 3.** JSA (Ou & Song, 2020) is obtained by defining

$$P_\lambda^k (\eta, d\eta') = K_\lambda^{N(k-1)+1} (\mathbf{z}^{(1)}, d\mathbf{z'}^{(1)}) \, K_\lambda^{N(k-1)+2} (\mathbf{z}^{(2)}, d\mathbf{z'}^{(2)}) \cdot \dots \cdot K_\lambda^{N(k-1)+N} (\mathbf{z}^{(N)}, d\mathbf{z'}^{(N)})$$

with $\eta_t = (\mathbf{z}_t^{(1)}, \mathbf{z}_t^{(2)}, \dots, \mathbf{z}_t^{(N)})$. Then, given Assumption 2 and 3, the mixing rate and the gradient variance bounds are

$$d_{\mathrm{TV}} (P_\lambda^k (\eta, \cdot), \Pi) \le C(r, N) \, r^{kN} \quad \text{and} \quad \mathbb{E} \left[ \|\mathbf{g}_{t,\mathrm{JSA}}\|_2^2 \,\middle|\, \mathcal{F}_{t-1} \right] \le L^2 \left[ \frac{1}{2} + \frac{3}{2} \frac{1}{N} + \mathcal{O}(1/w^* + r^{tN}) \right] + C_{\mathrm{cov}} + \|\boldsymbol{\mu}\|_2^2,$$

where $\boldsymbol{\mu} = \mathbb{E}_\pi \mathbf{s}(\lambda; \mathbf{z})$, $C_{\mathrm{cov}} = \frac{2}{N^2} \sum_{n=2}^{N} \sum_{m=1}^{n-1} \mathrm{Cov} \left( \mathbf{s}(\lambda; \mathbf{z}_t^{(n)}), \mathbf{s}(\lambda; \mathbf{z}_t^{(m)}) \,\middle|\, \mathcal{F}_{t-1} \right)$ is the sum of the covariance between the samples, $w^* = \sup_{\mathbf{z}} \pi(\mathbf{z})/q_{\mathrm{def.}}(\mathbf{z}; \lambda)$, and $C(r, N) > 0$ is a finite constant.

*Proof.* See the *full proof* in page 24.

**Discussion**    As shown in Theorem 3, JSA benefits from increasing $N$ in terms of a faster mixing rate. However, under lack of convergence and model misspecification (large $w^*$), the variance improvement becomes marginal. Specifically, in the large $w^*$ regime, the variance reduction is limited by the constant $1/2$ term. This is true even when, ideally, the covariance between the samples is ignorable such that $C_{\mathrm{cov}} \approx 0$. In practice, however, the covariance term $C_{\mathrm{cov}}$ will be positive, only increasing variance. Therefore, *in the large $w^*$ regime, JSA will perform poorly, and the variance reduction by increasing $N$ is fundamentally limited.*

## 5    Parallel Markov Chain Score Ascent

Our analysis in Section 4 suggests that the statistical performance of MSC, MSC-RB, and JSA are heavily affected by model specification and the state of convergence through $w^*$. Furthermore, for JSA, a large $w^*$ abolishes our ability to counterbalance the inefficiency by increasing the computational budget $N$. However, $\rho$ and $G$ do not equally impact convergence; recent results on MCGD suggest that gradient variance is more critical than the mixing rate (see Section 3.1). We turn to leverage this understanding to overcome the limitations of previous methods.

---

**Algorithm 1:** pMCSA

**Input:** initial samples $\mathbf{z}_0^{(1)}, \dots, \mathbf{z}_0^{(N)}$,
        initial parameter $\lambda_0$,
        number of iterations $T$,
        stepsize schedule $\gamma_t$

**for** $t = 1, 2, \dots, T$ **do**
     **for** $n = 1, 2, \dots, N$ **do**
        $\mathbf{z}_t^{(n)} \sim K_{\lambda_{t-1}}(\mathbf{z}_{t-1}^{(n)}, \cdot)$
     **end**
     $\mathbf{g}(\lambda) = -\frac{1}{N} \sum_{n=1}^{N} \mathbf{s}(\lambda; \mathbf{z}_t^{(n)})$
     $\lambda_t = \lambda_{t-1} - \gamma_t \, \mathbf{g}(\lambda_{t-1})$
**end**

---

### 5.1    Parallel Markov Chain Score Ascent

We propose a novel scheme, *parallel Markov chain score ascent* (pMCSA, Algorithm 1), that embraces a slower mixing rate in order to consistently achieve an $\mathcal{O}(1/N)$ variance reduction, even on challenging problems with a large $w^*$,

**Algorithm Description**    Unlike JSA that uses *$N$ sequential* Markov chain states, pMCSA operates *$N$ parallel* Markov chains. To maintain a similar per-iteration cost with JSA, it performs only a single Markov chain transition for each chain. Since the chains are independent, the Metropolis-Hastings rejections do not affect the variance of pMCSA.

**Theorem 4.** pMCSA, our proposed scheme, is obtained by setting

$$P_\lambda^k (\eta, d\eta') = K_\lambda^k (\mathbf{z}^{(1)}, d\mathbf{z'}^{(1)}) \, K_\lambda^k (\mathbf{z}^{(2)}, d\mathbf{z'}^{(2)}) \cdot \dots \cdot K_\lambda^k (\mathbf{z}^{(N)}, d\mathbf{z'}^{(N)})$$

with $\eta = (\mathbf{z}^{(1)}, \mathbf{z}^{(2)}, \dots, \mathbf{z}^{(N)})$. Then, given Assumption 2 and 3, the mixing rate and the gradient variance bounds are

$$d_{\mathrm{TV}} (P_\lambda^k (\eta, \cdot), \Pi) \le C(N) \, r^k \quad \text{and} \quad \mathbb{E} \left[ \|\mathbf{g}_{t,\mathrm{pMCSA}}\|_2^2 \,\middle|\, \mathcal{F}_{t-1} \right] \le L^2 \left[ \frac{1}{N} + \frac{1}{N} \left( 1 - \frac{1}{w^*} \right) \right] + \mathcal{O}(r^t) + \|\boldsymbol{\mu}\|_2^2,$$

where $\boldsymbol{\mu} = \mathbb{E}_\pi \mathbf{s}(\lambda; \mathbf{z})$, $w^* = \sup_{\mathbf{z}} \pi(\mathbf{z})/q_{\mathrm{def.}}(\mathbf{z}; \lambda)$ and $C(N) > 0$ is a finite constant.

*Proof.* See the *full proof* in page 27.

**Discussion**    Unlike JSA and MSC, the variance reduction rate of pMCSA is independent of $w^*$. Therefore, it should perform significantly better on challenging practical problems. If we consider the rate of Duchi *et al.* (2012), the combined rate is constant with respect to $N$ since it cancels out. In practice, however, we observe that increasing $N$ accelerates convergence quite dramatically. Therefore, the mixing rate independent convergence rates by Doan *et al.* (2020a,b) appears to better reflect practical performance. This is because **(i)** the mixing rate $\rho$ is a conservative *global* bound and **(ii)** the mixing rate will improve naturally as MCSA converges.

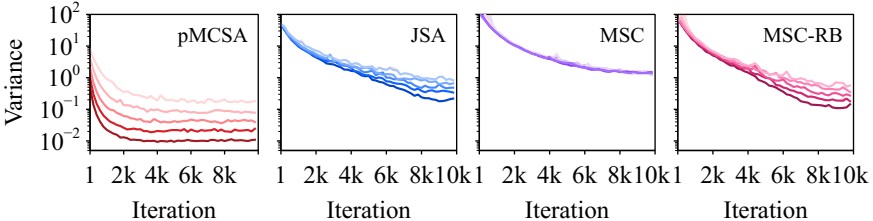

Figure 1: **Gradient variance versus iteration and computational budget ($N$). pMCSA not only achieves the least gradient variance, but its variance also scales better with $N$.** The colors range from light ($N = 2^3$) to dark ($N = 2^7$) representing the computational budgets $N \in [2^3, 2^4, 2^5, 2^6, 2^7]$. The target distribution is a 50-D multivariate Gaussian with $\nu = 500$. The error bands are the 80% quantiles obtained from 8 replications.

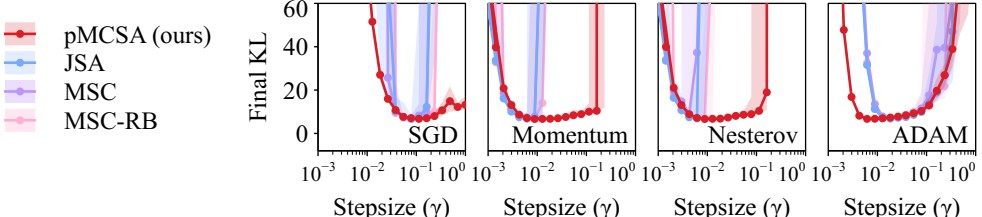

Figure 2: **Optimizer stepsize ($\gamma$) versus final KL. pMCSA is the least sensitive to optimizer hyperparameters and results in stable convergence.** The final KL is obtained at the $10^4$th iteration. The target distribution is a 100-D Gaussian with $\nu = 500$. The error bands are the 80% quantiles, while the solid lines are the median of 20 replications.

## 5.2 Computational Cost Comparison

The four schemes using the CIS and IMH kernels have different costs depending on $N$ as organized in Table 2.

**Cost of Sampling Proposals** For the CIS kernel used by MSC, $N$ controls the number of internal proposals sampled from $q(\cdot; \lambda)$. For JSA and pMCSA, the IMH kernel only uses a single sample from $q(\cdot; \lambda)$, but applies the kernel $N$ times. On the other hand, pMCSA needs twice more evaluations of $q(\cdot; \lambda)$. However, this added cost is minimal since it is dominated by that of evaluating $p(\mathbf{z}, \mathbf{x})$.

Table 2: Computational Costs

| | Applying $P_\lambda$ | | | Estimating $\mathbf{g}$ | |
|---|---|---|---|---|---|
| | $p(\mathbf{z}, \mathbf{x})$ # Eval. | $q(\mathbf{z}; \lambda)$ # Eval. | $q(\mathbf{z}; \lambda)$ # Samples | $p(\mathbf{z}, \mathbf{x})$ # Grad. | $q(\mathbf{z}; \lambda)$ # Grad. |
| ELBO | 0 | 0 | $N$ | $N$ | $N$ |
| MSC | $N-1$ | $N$ | $N-1$ | 0 | 1 |
| MSC-RB | $N-1$ | $N$ | $N-1$ | 0 | $N$ |
| JSA | $N$ | $N+1$ | $N$ | 0 | $N$ |
| **pMCSA** | $N$ | $2N$ | $N$ | 0 | $N$ |

**Cost of Estimating the Score** When estimating the score, MSC computes $\nabla_\lambda \log q(\mathbf{z}; \lambda)$ only once, while JSA and our proposed scheme compute it $N$ times. However, MSC-RB also computes the score $N$ times. Lastly, notice that MCSA methods do not differentiate through the likelihood $p(\mathbf{z}, \mathbf{x})$, unlike ELBO maximization, making its per-iteration cost significantly cheaper.

## 6 Evaluations

### 6.1 Experimental Setup

**Implementation** For the realistic experiments, we implemented[4] MCSA methods on top of the Turing (Ge *et al.*, 2018) probabilistic programming framework. For the variational family, we use diagonal multivariate Gaussians (mean-field family) with the support transformation of Kucukelbir *et al.* (2017). We use the ADAM optimizer by Kingma & Ba (2015) with a stepsize of 0.01 in all experiments. The budget is set to $N = 10$ for all experiments unless specified.

**Baselines** We compare **(i) pMCSA** (ours, Section 5), **(ii) JSA** (Ou & Song, 2020), **(iii) MSC** (Naesseth *et al.*, 2020), **(iv)** MSC with with Rao-Blackwellization (**MSC-RB**, Naesseth *et al.* 2020), and **(v)** evidence lower-bound maximization (**ELBO**, Kucukelbir *et al.* 2017; Ranganath *et al.* 2014) with the path derivative estimator (Roeder *et al.*, 2017).

---

[4] Available at `https://github.com/Red-Portal/KLpqVI.jl`

Table 3: Test Log Predictive Density on **Bayesian Neural Network Regression**

| | $D_\lambda$ | $D_{\mathbf{x}}$ | $N_{\text{train}}$ | ELBO | | MCSA Variants | | | |
| --- | --- | --- | --- | --- | --- | --- | --- | --- | --- |
| | | | | $N = 1$ | $N = 10$ | **pMCSA (ours)** | JSA | MSC | MSC-RB |
| yacht | 403 | 6 | 277 | **-2.45** ±0.01 | **-2.44** ±0.01 | **-2.49** ±0.01 | -3.00 ±0.05 | -2.98 ±0.04 | -2.98 ±0.04 |
| concrete | 503 | 8 | 927 | -3.25 ±0.01 | **-3.24** ±0.01 | **-3.20** ±0.01 | -3.33 ±0.02 | -3.32 ±0.02 | -3.33 ±0.02 |
| airfoil | 353 | 6 | 1352 | -2.53 ±0.02 | -2.56 ±0.02 | **-2.27** ±0.02 | -2.51 ±0.02 | -2.53 ±0.01 | -2.51 ±0.01 |
| energy | 503 | 9 | 691 | -2.42 ±0.02 | -2.40 ±0.02 | **-1.92** ±0.03 | -2.38 ±0.02 | -2.37 ±0.02 | -2.36 ±0.02 |
| wine | 653 | 12 | 1439 | **-0.96** ±0.01 | **-0.96** ±0.01 | **-0.95** ±0.01 | **-0.97** ±0.01 | **-0.97** ±0.01 | **-0.97** ±0.01 |
| boston | 753 | 14 | 455 | -2.72 ±0.03 | **-2.70** ±0.03 | **-2.69** ±0.02 | -2.82 ±0.02 | -2.80 ±0.03 | -2.78 ±0.02 |
| sml | 1203 | 23 | 3723 | -1.32 ±0.01 | **-1.25** ±0.02 | **-1.22** ±0.01 | -1.72 ±0.01 | -1.97 ±0.02 | -1.95 ±0.02 |
| gas | 6503 | 129 | 2308 | -0.06 ±0.01 | **0.13** ±0.03 | -0.09 ±0.02 | -0.47 ±0.03 | -0.47 ±0.04 | -0.50 ±0.03 |

[1] $D_\lambda$: Dimentionality of $\lambda$, $D_{\mathbf{x}}$: Number of features, $N_{\text{train}}$: Number of training data points.
[2] ± denotes the 95% bootstrap confidence intervals obtained from 20 replications.
[3] Bolded numbers don't have enough evidence to be distinguished from the best performing method under a .05 significance threshold (Friedman test with Nemenyi post-hoc test, Demšar 2006).

## 6.2 Simulations

**Setup** First, we verify our theoretical analysis on multivariate Gaussians with full-rank covariances sampled from Wishart distribution with $\nu$ degrees of freedom (values of $\nu$ are in the figure captions). This problem is challenging since an IMH (used by pMCSA, JSA) or CIS (used by MSC, MSC-RB) kernel with a diagonal Gaussian proposal will mix slowly due to a large $w^*$.

**Gradient Variance** We evaluate our theoretical analysis of the gradient variance. The variance is estimated from 512 independent Markov chains using the parameters generated by the main MCSA procedure. The estimated variances are shown in Figure 1. We make the following observations: **(i)** pMCSA has the lowest variance overall, and it consistently benefits from increasing $N$. **(ii)** MSC does not benefit from increasing $N$ whatsoever. **(iii)** MSC-RB does not benefit much from increasing $N$ until the $\chi^2$ divergence between $\pi$ and $q(\cdot; \lambda)$ has become small. **(iv)** JSA does not benefit from increasing $N$ until $q(\cdot; \lambda)$ has sufficiently converged (when $w^*$ has become small). These results confirm our theoretical analyses in Sections 4 and 5.

**Robustness Against Optimizers** Since the convergence of most sophisticated SGD optimizers has yet to be established for MCGD, we empirically investigate their effectiveness. The results using SGD (Bottou *et al.*, 2018; Robbins & Monro, 1951), Momentum (Polyak, 1964), Nesterov (Nesterov, 1983), ADAM (Kingma & Ba, 2015), and varying stepsizes are shown in Figure 2. Clearly, pMCSA successfully converges for the broadest variety of optimizer settings. Overall, most MCSA methods seem to be the most stable with ADAM, which points out that establishing the convergence of ADAM for MCGD will be a promising direction for future works.

## 6.3 Bayesian Neural Network Regression

**Setup** For realistic experiments, we train Bayesian neural networks (BNN, Neal 1996) for regression. We use datasets from the UCI repository (Dua & Graff, 2017) with 90% random train-test splits and run for $T = 5 \cdot 10^4$ iterations. We use the model and forward propagation method of Hernandez-Lobato & Adams (2015) with a 50-unit hidden layer (see Appendix C.1).

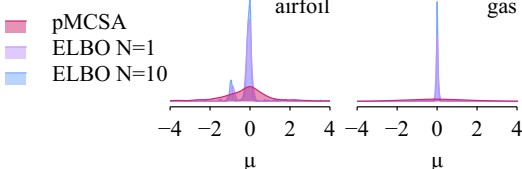

Figure 3: **Distribution of the variational posterior mean of the BNN weights. pMCSA results in much less pruning.** The density was estimated with a Gaussian kernel and the bandwidth was selected with Silverman's rule.

**Results** The results are shown in Table 3. pMCSA achieves the best performance compared to all other MCSA methods. Also, its overall performance is comparable to exclusive KL minimization methods (ELBO) unlike other MCSA methods. Furthermore, on airfoil and energy, pMCSA improves over ELBO by 0.29 nat and 0.48 nat. Even on gas where pMCSA did not beat ELBO, its performance is comparable, and it dominates all other MCSA methods by roughly 0.4 nat. Additional experimental results, including root mean-square error (RMSE) results and plots with respect to the wall clock time, can be found in Appendix E.1

Table 4: Test Log Predictive Density on **Robust Gaussian Process Regression**

| | $D_\lambda$ | $D_\mathbf{x}$ | $N_{\text{train}}$ | ELBO $N = 1$ | MCSA Variants pMCSA (ours) | JSA | MSC | MSC-RB |
|---|---|---|---|---|---|---|---|---|
| yacht | 287 | 6 | 277 | -3.63 ±0.02 | **-3.31** ±0.04 | **-3.29** ±0.05 | **-3.25** ±0.04 | **-3.27** ±0.05 |
| airfoil | 353 | 6 | 1352 | -3.14 ±0.01 | **-2.63** ±0.01 | -2.83 ±0.04 | -2.77 ±0.02 | **-2.73** ±0.02 |
| boston | 472 | 13 | 455 | **-2.98** ±0.01 | **-2.96** ±0.02 | **-3.00** ±0.03 | **-3.00** ±0.03 | **-2.96** ±0.03 |
| energy | 703 | 8 | 691 | -2.75 ±0.01 | **-2.58** ±0.03 | -2.78 ±0.04 | -2.70 ±0.04 | -2.72 ±0.05 |
| concrete | 939 | 8 | 927 | -3.68 ±0.01 | **-3.49** ±0.01 | -3.69 ±0.02 | **-3.59** ±0.04 | **-3.57** ±0.02 |
| wine | 1454 | 11 | 1439 | -1.02 ±0.01 | **-0.94** ±0.02 | -1.04 ±0.01 | -1.00 ±0.02 | -0.99 ±0.02 |
| gas | 2440 | 128 | 2308 | **0.18** ±0.02 | **-0.86** ±0.02 | -1.10 ±0.03 | -1.10 ±0.04 | -1.06 ±0.02 |

[1] $D_\lambda$: Dimentionality of $\lambda$, $D_\mathbf{x}$: Number of features, $N_{\text{train}}$: Number of training data points.

[2] ± denotes the 95% bootstrap confidence intervals obtained from 20 replications.

[3] Bolded numbers don't have enough evidence to be distinguished from the best performing method under a .05 significance threshold (Friedman test with Nemenyi post-hoc test, Demšar 2006).

**Weight Pruning**    When using VI, BNNs have been known to underfit data, which Hoffman (2017); MacKay (2001); Trippe & Turner (2017) associated with "weight pruning." That is, the variational posterior converges to the zero-mean prior. Furthermore, Coker et al. (2022); Huix et al. (2022) have shown that this is guaranteed to happen under certain conditions. However, these results are strictly based on exclusive KL minimization. On the other hand, Figure 3 shows that pMCSA does not suffer from weight pruning, which suggests that pruning is an artifact of using the exclusive KL. Additional plots are shown in Figure 6 (Appendix E.1).

### 6.4    Robust Gaussian Process Regression

**Setup**    We train Gaussian processes (GP) with a Student-T likelihood for robust regression. We use datasets from the UCI repository (Dua & Graff, 2017) with 90% random train-test splits. We use the Matérn 5/2 covariance kernel with automatic relevance determination (Neal, 1996) (see Appendix C.2). We run all methods with $T = 2 \cdot 10^4$ iterations. For prediction, we use the mode of $q(\cdot; \lambda)$ for the hyperparameters and marginalize the latent function over $q(\cdot; \lambda)$ (Rasmussen & Williams, 2006). We consider ELBO with only $N = 1$ since differentiating through the likelihood makes its per-iteration cost comparable to MCSA methods with $N = 10$.

**Results**    The results are shown in Table 4. Except for gas, pMCSA achieves better predictive densities than all other methods. This suggests that, overall, the exclusive KL may be less effective in terms of uncertainty quantification for GP posteriors. While ELBO achieves the best performance on gas, among MCSA methods, pMCSA dominates. Our encouraging regression results suggest that incorporating methods such as inducing points (Snelson & Ghahramani, 2005) into MCSA may lead to an important new class of GP models. Additional experimental results, including RMSE results and plots with respect to the wall clock time, can be found in Appendix E.2. Note that in Appendix E.2, MCSA methods appear worse in terms of RMSE compared to the exclusive KL since the inclusive KL puts less probability volume around the posterior mode.

## 7    Related Works

**Inclusive KL minimization**    Our MCSA framework generalizes MSC (Naesseth et al., 2020) and JSA Ou & Song (2020), which are inclusive KL minimization based on SGD and Markov chains. Similar to MCSA is the method of Li et al. (2017). However, the convergence of this method is not guaranteed since it uses short Markov chains, disqualifying for MCSA. Other methods based on biased gradients have been proposed by Bornschein & Bengio (2015); Le et al. (2020), but these are specific for deep generative models. On a different note, Jerfel et al. (2021) use boosting instead of SGD to minimize the inclusive KL, which gradually builds a complex variational approximation from a simple variational family.

**Beyond the KL Divergence**    Discovering alternative divergences for VI has been an active research area. For example, the $\chi^2$ (Dieng et al., 2017), $f$ (Wan et al., 2020; Wang et al., 2018), $\alpha$ (Hernandez-Lobato et al., 2016; Li & Turner, 2016; Regli & Silva, 2018), reguarlized importance ratio (Bamler et al., 2017) divergences have been studied for VI. However, for gradient estimation these methods involve the importance ratio $w(\mathbf{z}) = \pi(\mathbf{z})/q(\mathbf{z})$, which leads to significant variance and low signal-to-noise ratio under model misspecification (Bamler et al., 2017;

Geffner & Domke, 2021a,c). In contrast, under stationarity, the variance of pMCSA is $\sigma^2/N$ ($\sigma$ is the variance of the score over the posterior) regardless of model misspecification. Meanwhile, Geffner & Domke (2021b); Ruiz & Titsias (2019); Salimans *et al.* (2015); Zhang *et al.* (2021) construct implicit divergences formed by MCMC. With the exception of Ruiz & Titsias (2019), most of these approaches aim to maximize auxiliary representations of the classic ELBO. Therefore, their property is likely to be similar to the exclusive KL.

**Adaptive MCMC and MCSA** As pointed out by Ou & Song (2020), using $q(\cdot; \lambda)$ within the MCMC kernel makes MCSA structurally equivalent to adaptive MCMC. In particular, Andrieu & Thoms (2008); Brofos *et al.* (2022); Gabrié *et al.* (2022); Garthwaite *et al.* (2016) discuss the use of stochastic approximation in adaptive MCMC. Also, Andrieu & Moulines (2006); Brofos *et al.* (2022); Giordani & Kohn (2010); Habib & Barber (2019); Holden *et al.* (2009); Keith *et al.* (2008); Neklyudov *et al.* (2019) specifically discuss adapting the propsosal of IMH kernels, and some of them use KL divergence minimization. These methods focus on showing ergodicity the samples ($\eta_t$ in our context) not the convergence of the variational approximation $q(\cdot; \lambda)$. In this work, we focused on the convergence of $q(\cdot; \lambda)$, which could advance the adaptive MCMC side of the story.

# 8 Discussions

This paper presented a new theoretical framework for analyzing inclusive KL divergence minimization methods based on running SGD with Markov chains. Furthermore, we proposed pMCSA, a new MCSA method that enjoys substantially low variance. We have shown that this theoretical improvement translates into better empirical performance.

**Limitations** Our work has three main limitations. Firstly, since our work aims to understand existing MCSA methods, it inherits their current limitations. For example, minibatch subsampling is challenging for models with non-factorizable likelihoods (Naesseth *et al.*, 2020). Secondly, our theoretical analysis in Section 4 requires Assumption 3, which is strong, but required to connect with MCGD. An important future direction would be to relax the assumptions needed by MCGD. Lastly, our MCSA framework does not include models with parameterized posteriors such as variational autoencoders.

**Parameterized Posteriors** On problems with parameterized posteriors, the target posterior moves around. Therefore, quickly chasing the moving posterior with fast converging MCMC kernels is as important as achieving low variance. Because of this, trading bias and variance is less straightforward compared to the "static" setting we consider. Furthermore, the usage of expensive MCMC kernels could be beneficial as suggested by Zhang *et al.* (2022).

**Towards Alternative Divergences** In Section 6, we have shown that minimizing the *inclusive* KL is competitive against minimizing the *exclusive* KL on general Bayesian inference problems. Although Dhaka *et al.* (2021) have shown that the inclusive KL fails on high-dimensional problems, this is only the case under the presence of strong correlations. Before entirely ditching the inclusive KL, it is essential to ask, "how correlated are posteriors really in practice?" Furthermore, the true performance of alternative divergences is often masked by the limitations of the inference procedure (Geffner & Domke, 2021a,c). Given that pMCSA significantly advances the best-known performance of inclusive KL minimization, it is possible that similar improvements could be extracted from other divergences. To conclude, our results motivate further development of better inference algorithms for alternative divergence measures.

## Acknowledgments and Disclosure of Funding

We thank Hongseok Yang for pointing us to relevant related work, Guanyang Wang for insightful discussions about the independent Metropolis-Hastings algorithm, Geon Park and Kwanghee Choi for constructive comments that enriched this paper, Christian A. Naesseth for comments about Rao-Blackwellized MSC, and Gabriel V. Cardoso for comments about BR-SNIS. We also acknowledge the Department of Computer Science and Engineering of Sogang University for providing computational resources.

K. Kim was supported in part by the EPSRC through the Big Hypotheses grant [EP/R018537/1]. J. R. Gardner was supported by NSF award [IIS-2145644]. H. Kim was supported by the Basic Science Research Program through the National Research Foundation of Korea (NRF) funded by the Ministry of Science and ICT under Grant [NRF-2021R1A2C1095435].

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
