# APPENDIX OF
# MARKOV CHAIN SCORE ASCENT:
# A Unifying Framework *of*
# Variational Inference *with* Markovian Gradients

## A   Computational Resources

Table 5: Computational Resources for Bayesian Neural Network Regression

| Type | Model and Specifications |
|---|---|
| System Topology | 4 nodes with 20 logical threads each |
| Processor | Intel Xeon Xeon E5–2640 v4, 2.2 GHz (maximum 3.1 GHz) |
| Cache | 32 kB L1, 256 kB L2, and 25 MB L3 |
| Memory | 64GB RAM |

Table 6: Computational Resources for Robust Gaussian Process Regression

| Type | Model and Specifications |
|---|---|
| System Topology | 1 node with 16 logical threads |
| Processor | AMD EPYC 7262, 3.2 GHz (maximum 3.4 GHz) |
| Accelerator | NVIDIA Titan RTX, 1.3 GHZ, 24GB RAM |
| Cache | 256 kB L1, 4MiB L2, and 128MiB L3 |
| Memory | 126GB RAM |

## B   Pseudocodes

### B.1   Markov Chain Monte Carlo Kernels

---
**Algorithm 2:** Conditional Importance Sampling Kernel

---
**Input:** previous sample $\mathbf{z}_{t-1}$,
      previous parameter $\lambda_{t-1}$,
      number of proposals $N$

$\mathbf{z}^{(0)} = \mathbf{z}_{t-1}$

$\mathbf{z}^{(i)} \sim q_{\text{def.}}\left(\mathbf{z}; \lambda_{t-1}\right) \quad$ for $i = 1, 2, \ldots, N$

$\widetilde{w}(\mathbf{z}^{(i)}) = p(\mathbf{z}^{(i)}, \mathbf{x}) / q_{\text{def.}}\left(\mathbf{z}^{(i)}; \lambda_{t-1}\right) \quad$ for $i = 0, 1, \ldots, N$

$\overline{w}^{(i)} = \dfrac{\widetilde{w}(\mathbf{z}^{(i)})}{\sum_{i=0}^{N} \widetilde{w}(\mathbf{z}^{(i)})} \quad$ for $i = 0, 1, \ldots, N$

$\mathbf{z}_t \sim \text{Multinomial}(\overline{w}^{(0)}, \overline{w}^{(1)}, \ldots, \overline{w}^{(N)})$

---

---

**Algorithm 3:** Independent Metropolis-Hastings Kernel

---

**Input:** previous sample $\mathbf{z}_{t-1}$,
   previous parameter $\boldsymbol{\lambda}_{t-1}$,

$\mathbf{z}^* \sim q_{\text{def.}}(\mathbf{z}; \boldsymbol{\lambda}_{t-1})$
$\widetilde{w}(\mathbf{z}) = p(\mathbf{z}, \mathbf{x})/q_{\text{def.}}(\mathbf{z}; \boldsymbol{\lambda}_{t-1})$
$\alpha = \min(\widetilde{w}(\mathbf{z}^*)/\widetilde{w}(\mathbf{z}_{t-1}), 1)$
$u \sim \text{Uniform}(0, 1)$
**if** $u < \alpha$ **then**
 |  $\mathbf{z}_t = \mathbf{z}^*$
**else**
 |  $\mathbf{z}_t = \mathbf{z}_{t-1}$
**end**

---

## B.2 Markov Chain Score Ascent Algorithms

---

**Algorithm 4:** Markovian Score Climbing

---

**Input:** Initial sample $\mathbf{z}_0$,
   initial parameter $\boldsymbol{\lambda}_0$,
   number of iterations $T$,
   stepsize schedule $\gamma_t$
**for** $t = 1, 2, \ldots, T$ **do**
 |  $\mathbf{z}_t \sim K_{\boldsymbol{\lambda}_{t-1}}(\mathbf{z}_{t-1}, \cdot)$
 |  $\mathbf{g}(\boldsymbol{\lambda}) = -\mathbf{s}(\boldsymbol{\lambda}; \mathbf{z}_t)$
 |  $\boldsymbol{\lambda}_t = \boldsymbol{\lambda}_{t-1} - \gamma_t \mathbf{g}(\boldsymbol{\lambda}_{t-1})$
**end**

---

---

**Algorithm 5:** Joint Stochastic Approximation

---

**Input:** Initial sample $\mathbf{z}_0^{(N)}$,
   initial parameter $\boldsymbol{\lambda}_0$,
   number of iterations $T$,
   stepsize schedule $\gamma_t$
**for** $t = 1, 2, \ldots, T$ **do**
 |  $\mathbf{z}_t^{(0)} = \mathbf{z}_{t-1}^{(N)}$
 |  **for** $n = 1, 2, \ldots, N$ **do**
 |  |  $\mathbf{z}_t^{(n)} \sim K_{\boldsymbol{\lambda}_{t-1}}(\mathbf{z}_t^{(n-1)}, \cdot)$
 |  **end**
 |  $\mathbf{g}(\boldsymbol{\lambda}) = -\frac{1}{N}\sum_{n=1}^{N}\mathbf{s}(\boldsymbol{\lambda}; \mathbf{z}_t^{(n)})$
 |  $\boldsymbol{\lambda}_t = \boldsymbol{\lambda}_{t-1} - \gamma_t \mathbf{g}(\boldsymbol{\lambda}_{t-1})$
**end**

---

## C  Probabilistic Models Used in the Experiments

### C.1  Bayesian Neural Network Regression

We use the BNN model of Hernandez-Lobato & Adams (2015) defined as

$$
\begin{aligned}
\lambda^{-1} &\sim \text{inverse-gamma} \left( \alpha = 6, \beta = 6 \right) \\
\gamma^{-1} &\sim \text{inverse-gamma} \left( \alpha = 6, \beta = 6 \right) \\
\mathbf{W}_1 &\sim \mathcal{N} \left( \mathbf{0}, \lambda^{-1} \mathbf{I} \right) \\
\mathbf{z} &= \text{ReLU} \left( \mathbf{W}_1 \mathbf{x}_i \right) \\
\mathbf{W}_2 &\sim \mathcal{N} \left( \mathbf{0}, \lambda^{-1} \mathbf{I} \right) \\
\hat{y} &= \text{ReLU} \left( \mathbf{W}_2 \mathbf{z} \right) \\
y_i &\sim \mathcal{N} \left( \hat{y}, \gamma^{-1} \right),
\end{aligned}
$$

where $\mathbf{x}_i$ and $y_i$ are the feature vector and target value of the $i$th datapoint. Given the variational distribution of $\lambda^{-1}, \gamma^{-1}, \mathbf{W}_1, \mathbf{W}_2$, we use the same posterior predictive approximation of Hernandez-Lobato & Adams (2015). We apply z-standardization (whitening) to the features $\mathbf{x}_i$ and the target values $y_i$, and unwhiten the predictive distribution.

### C.2  Robust Gaussian Process Logistic Regression

We perform robust Gaussian process regression by using a student-t prior with a latent Gaussian process prior. The model is defined as

$$
\begin{aligned}
\log \sigma_f &\sim \mathcal{N}(0, 4) \\
\log \epsilon &\sim \mathcal{N}(0, 4) \\
\log \ell_i &\sim \mathcal{N}(0, 0.2) \\
f &\sim \mathcal{GP} \left( \mathbf{0}, \mathbf{\Sigma}_{\sigma_f, \ell} + \left( \delta + \epsilon^2 \right) \mathbf{I} \right) \\
\nu &\sim \text{gamma} \left( \alpha = 4, \beta = 1/10 \right) \\
\log \sigma_y &\sim \mathcal{N}(0, 4) \\
y_i &\sim \text{student-t} \left( f \left( \mathbf{x}_i \right), \sigma_y, \nu \right).
\end{aligned}
$$

The covariance $\mathbf{\Sigma}$ is computed using a kernel $k \left( \cdot, \cdot \right)$ such that $[\mathbf{\Sigma}]_{i,j} = k \left( \mathbf{x}_i, \mathbf{x}_j \right)$ where $\mathbf{x}_i$ and $\mathbf{x}_j$ are data points in the dataset. For the kernel, we use the Matern 5/2 kernel with automatic relevance determination (Neal, 1996) defined as

$$
k \left( \mathbf{x}, \mathbf{x}'; \ \sigma^2, \ell_1^2, \dots, \ell_D^2 \right) = \sigma_f \left( 1 + \sqrt{5}r + \frac{5}{3}r^2 \right) \exp \left( -\sqrt{5}r \right), \quad \text{where} \quad r = \sum_{i=1}^{D} \frac{\left( \mathbf{x}_i - \mathbf{x}_i' \right)^2}{\ell_i^2}
$$

and $D$ is the number of dimensions. The jitter term $\delta$ is used for numerical stability. We set a small value of $\delta = 1 \times 10^{-6}$.

# D Proofs

**Proposition 1.** Let $\eta = \left(\mathbf{z}^{(1)}, \mathbf{z}^{(2)}, \dots, \mathbf{z}^{(N)}\right)$ and a Markov chain kernel $P_\lambda(\eta, \cdot)$ be $\Pi$-invariant where $\Pi$ is defined as

$$\Pi(\eta) = \pi\left(\mathbf{z}^{(1)}\right) \pi\left(\mathbf{z}^{(2)}\right) \times \dots \times \pi\left(\mathbf{z}^{(N)}\right).$$

Then, by defining the objective function $f$ and the gradient estimator $\mathbf{g}$ to be

$$f(\lambda, \eta) = -\frac{1}{N} \sum_{n=1}^{N} \log q\left(\mathbf{z}^{(n)}; \lambda\right) - \mathbb{H}[\pi] \quad \text{and} \quad \mathbf{g}(\lambda, \eta) = -\frac{1}{N} \sum_{n=1}^{N} \mathbf{s}\left(\mathbf{z}^{(n)}; \lambda\right),$$

where $\mathbb{H}[\pi]$ is the entropy of $\pi$, MCGD results in inclusive KL minimization as

$$\mathbb{E}_\Pi\left[f(\lambda, \eta)\right] = d_{\mathrm{KL}}(\pi \parallel q(\cdot; \lambda)) \quad \text{and} \quad \mathbb{E}_\Pi\left[\mathbf{g}(\lambda, \eta)\right] = \nabla_\lambda d_{\mathrm{KL}}(\pi \parallel q(\cdot; \lambda)).$$

*Proof.* For notational convenience, we define the shorthand

$$\pi\left(\mathbf{z}^{(1:N)}\right) = \pi\left(\mathbf{z}^{(1)}\right) \pi\left(\mathbf{z}^{(2)}\right) \times \dots \times \pi\left(\mathbf{z}^{(N)}\right).$$

Then,

$$\mathbb{E}_\Pi\left[f(\lambda, \eta)\right]$$

$$= \int \left(-\frac{1}{N} \sum_{n=1}^{N} \log q\left(\mathbf{z}^{(n)}; \lambda\right) - \mathbb{H}[\pi]\right) \pi\left(\mathbf{z}^{(1:N)}\right) d\mathbf{z}^{(1:N)}$$

$$= \int \left(-\frac{1}{N} \sum_{n=1}^{N} \log q\left(\mathbf{z}^{(n)}; \lambda\right)\right) \pi\left(\mathbf{z}^{(1:N)}\right) d\mathbf{z}^{(1:N)} - \mathbb{H}[\pi]$$

$$= \frac{1}{N} \sum_{n=1}^{N} \left\{ \int \left(-\log q\left(\mathbf{z}^{(n)}; \lambda\right)\right) \pi\left(\mathbf{z}^{(1:N)}\right) d\mathbf{z}^{(1:N)} \right\} - \mathbb{H}[\pi]$$

$$= \frac{1}{N} \sum_{n=1}^{N} \int \left(-\log q\left(\mathbf{z}^{(n)}; \lambda\right)\right) \pi\left(\mathbf{z}^{(n)}\right) d\mathbf{z}^{(n)} - \mathbb{H}[\pi] \qquad \textit{Marginalized } \mathbf{z}^{(m)} \textit{ for all } m \neq n$$

$$= \frac{1}{N} \sum_{n=1}^{N} \int \left(-\log q\left(\mathbf{z}^{(n)}; \lambda\right) + \log \pi\left(\mathbf{z}^{(n)}\right)\right) \pi\left(\mathbf{z}^{(n)}\right) d\mathbf{z}^{(n)} \qquad \textit{Definition of } \mathbb{H}[\pi]$$

$$= \frac{1}{N} \sum_{n=1}^{N} \int \pi\left(\mathbf{z}^{(n)}\right) \log \frac{\pi\left(\mathbf{z}^{(n)}\right)}{q\left(\mathbf{z}^{(n)}; \lambda\right)} d\mathbf{z}^{(n)}$$

$$= \frac{1}{N} \sum_{n=1}^{N} d_{\mathrm{KL}}(\pi \parallel q(\cdot; \lambda)) \qquad \textit{Definition of } d_{\mathrm{KL}}$$

$$= d_{\mathrm{KL}}(\pi \parallel q(\cdot; \lambda)). \tag{4}$$

For $\mathbb{E}_\Pi\left[\mathbf{g}(\lambda, \eta)\right]$, note that

$$\nabla_\lambda f(\lambda, \eta) = -\frac{1}{N} \sum_{n=1}^{N} \nabla_\lambda \log q\left(\mathbf{z}^{(n)}; \lambda\right) = -\frac{1}{N} \sum_{n=1}^{N} \mathbf{s}\left(\mathbf{z}^{(n)}; \lambda\right) = \mathbf{g}(\lambda, \eta). \tag{5}$$

Therefore, it suffices to show that

$$\begin{aligned} \nabla_\lambda d_{\mathrm{KL}}(\pi \parallel q(\cdot; \lambda)) &= \nabla_\lambda \mathbb{E}_\Pi\left[f(\lambda, \eta)\right] & \textit{Equation (4)} \\ &= \mathbb{E}_\Pi\left[\nabla_\lambda f(\lambda, \eta)\right] & \textit{Leibniz derivative rule} \\ &= \mathbb{E}_\Pi\left[\mathbf{g}(\lambda, \eta)\right]. & \textit{Equation (5)} \end{aligned}$$

$\square$

**Proposition 2.** The maximum importance weight $w^* = \sup_{\mathbf{z}} w(\mathbf{z}) = \sup_{\mathbf{z}} \pi(\mathbf{z})/q(\mathbf{z}; \lambda)$ is bounded below exponentially by the KL divergence as

$$\exp\left(d_{\mathrm{KL}}(\pi \parallel q(\cdot; \lambda))\right) < w^*.$$

*Proof.*

$$d_{\mathrm{KL}}(\pi \parallel q(\cdot;\lambda)) = \mathbb{E}_{\mathbf{z} \sim \pi(\cdot)}\left[\log \frac{\pi(\mathbf{z})}{q(\mathbf{z};\lambda)}\right] \quad \textit{Definition of } d_{KL}$$

$$\leq \log \mathbb{E}_{\mathbf{z} \sim \pi(\cdot)}\left[\frac{\pi(\mathbf{z})}{q(\mathbf{z};\lambda)}\right] \quad \textit{Jensen's inequality}$$

$$\leq \log \mathbb{E}_{\mathbf{z} \sim \pi(\cdot)}\left[w^*\right]$$

$$= \log w^*.$$

$\square$

**Lemma 1.** *For the probability measures $p_1, \dots, p_N$ and $q_1, \dots, q_N$ defined on a measurable space $(\mathsf{X}, \mathcal{A})$ and an arbitrary set $A \in \mathcal{A}$,*

$$\left|\int_{A^N} p_1(dx_1) p_2(dx_2) \times \dots \times p_N(dx_N) - q_1(dx_1) q_2(dx_2) \times \dots \times q_N(dx_N)\right|$$

$$\leq \sum_{n=1}^{N}\left|\int_A p_n(dx_n) - q_n(dx_n)\right|$$

*Proof.* By using the following shorthand notations

$$p_{(1:N)}\left(dx_{(1:N)}\right) = p_1(dx_1) p_2(dx_2) \times \dots \times p_N(dx_N)$$

$$q_{(1:N)}\left(dx_{(1:N)}\right) = q_1(dx_1) q_2(dx_2) \times \dots \times q_N(dx_N),$$

the result follows from induction as

$$\left|\int_{A^N} p_{(1:N)}\left(dx_{(1:N)}\right) - q_{(1:N)}\left(dx_{(1:N)}\right)\right|$$

$$= \left|\left(\int_A p_1(dx_1) - q_1(dx_1)\right)\int_{A^{N-1}} p_{(2:N)}\left(dx_{(2:N)}\right)\right.$$

$$\left. + \int_A q_1(dx_1)\left(\int_{A^{N-1}} p_{(2:N)}\left(dx_{(2:N)}\right) - q_{(2:N)}\left(dx_{(2:N)}\right)\right)\right|$$

$$\leq \left|\int_A p_1(dx_1) - q_1(dx_1)\right|\int_{A^{N-1}} p_{(2:N)}\left(dx_{(2:N)}\right)$$

$$+ \int_A q_1(dx_1)\left|\int_{A^{N-1}} p_{(2:N)}\left(dx_{(2:N)}\right) - q_{(2:N)}\left(dx_{(2:N)}\right)\right| \quad \textit{Triangle inequality}$$

$$\leq \left|\int_A p_1(dx_1) - q_1(dx_1)\right|$$

$$+ \left|\int_{A^{N-1}} p_{(2:N)}\left(dx_{(2:N)}\right) - q_{(2:N)}\left(dx_{(2:N)}\right)\right|. \quad \textit{Applied } p_n(A), q_n(A) \leq 1$$

$\square$

**Lemma 2.** *Let $\mathbf{g}$ be a vector-valued, biased estimator of $\boldsymbol{\mu}$, where the bias is denoted as $\mathrm{Bias}[\mathbf{g}] = \|\mathbb{E}\mathbf{g} - \boldsymbol{\mu}\|_2$ and the mean-squared error is denoted as $\mathrm{MSE}[\mathbf{g}] = \mathbb{E}\|\mathbf{g} - \boldsymbol{\mu}\|_2^2$. Then, the second moment of $\mathbf{g}$ is bounded as*

➊ $\mathbb{E}\|\mathbf{g}\|_2^2 \leq \mathbb{V}\mathbf{g} + \mathrm{Bias}[\mathbf{g}]^2 + 2\,\mathrm{Bias}[\mathbf{g}]\,\|\boldsymbol{\mu}\|_2 + \|\boldsymbol{\mu}\|_2^2,$

➋ $\mathbb{E}\|\mathbf{g}\|_2^2 \leq \mathrm{MSE}[\mathbf{g}] + 2\,\mathrm{Bias}[\mathbf{g}]\,\|\boldsymbol{\mu}\|_2 + \|\boldsymbol{\mu}\|_2^2,$

*where $\mathbb{V}\mathbf{g} = \mathbb{E}\|\mathbf{g} - \mathbb{E}\mathbf{g}\|_2^2$ is the variance of the estimator.*

*Proof.* ❶ follows from the decomposition

$$
\begin{aligned}
\mathbb{E}\left[\|\mathbf{g}\|_2^2\right] &= \mathbb{V}\mathbf{g} + \|\mathbb{E}\mathbf{g}\|_2^2 \\
&= \mathbb{V}\mathbf{g} + \|\mathbb{E}\mathbf{g} - \boldsymbol{\mu} + \boldsymbol{\mu}\|_2^2 \\
&= \mathbb{V}\mathbf{g} + \|\mathbb{E}\mathbf{g} - \boldsymbol{\mu}\|_2^2 + 2\left(\mathbb{E}\mathbf{g} - \boldsymbol{\mu}\right)^\top \boldsymbol{\mu} + \|\boldsymbol{\mu}\|_2^2 \qquad \textit{Expanded quadratic} \\
&\leq \mathbb{V}\mathbf{g} + \|\mathbb{E}\mathbf{g} - \boldsymbol{\mu}\|_2^2 + 2\|\mathbb{E}\mathbf{g} - \boldsymbol{\mu}\|_2\|\boldsymbol{\mu}\|_2 + \|\boldsymbol{\mu}\|_2^2 \qquad \textit{Cauchy-Shwarz inequality} \\
&= \mathbb{V}\mathbf{g} + \text{Bias}\left[\mathbf{g}\right]^2 + 2\,\text{Bias}\left[\mathbf{g}\right]\|\boldsymbol{\mu}\|_2 + \|\boldsymbol{\mu}\|_2^2. \qquad \textit{Definition of bias}
\end{aligned}
$$

Meanwhile, by the well-known bias-variance decomposition formula of the mean-squared error, ❷ directly follows from ❶. □

**Theorem 1.** MSC (Naesseth *et al.*, 2020) is obtained by defining

$$
P_\lambda^k\left(\boldsymbol{\eta}, d\boldsymbol{\eta}'\right) = K_\lambda^k\left(\mathbf{z}, d\mathbf{z}'\right)
$$

with $\boldsymbol{\eta}_t = \mathbf{z}_t$, where $K_\lambda\left(\mathbf{z}, \cdot\right)$ is the CIS kernel with $q_{\text{def.}}\left(\cdot; \lambda\right)$ as its proposal distribution. Then, given Assumption 2 and 3, the mixing rate and the gradient bounds are given as

$$
d_{\text{TV}}\left(P_\lambda^k\left(\boldsymbol{\eta}, \cdot\right), \Pi\right) \leq \left(1 - \frac{N-1}{2w^* + N - 2}\right)^k \quad \text{and} \quad \mathbb{E}\left[\|\mathbf{g}_{t,\text{MSC}}\|^2 \,\middle|\, \mathcal{F}_{t-1}\right] \leq L^2,
$$

where $w^* = \sup_{\mathbf{z}} \pi\left(\mathbf{z}\right) / q_{\text{def.}}\left(\mathbf{z}; \lambda\right)$.

*Proof.* MSC is described in Algorithm 4. At each iteration, it performs a single MCMC transition with the CIS kernel where it internally uses $N$ proposals. That is,

$$
\begin{aligned}
\mathbf{z}_t \mid \mathbf{z}_{t-1}, \boldsymbol{\lambda}_{t-1} &\sim K_{\boldsymbol{\lambda}_{t-1}}\left(\mathbf{z}_{t-1}, \cdot\right) \\
\mathbf{g}_{t,\text{MSC}} &= -\mathbf{s}\left(\lambda, \mathbf{z}_t\right),
\end{aligned}
$$

where $K_{\boldsymbol{\lambda}_{t-1}}$ is the CIS kernel using $q_{\text{def.}}\left(\cdot; \boldsymbol{\lambda}_{t-1}\right)$.

**Ergodicity of the Markov Chain**   The ergodic convergence rate of $P_\lambda$ is equal to that of $K_\lambda$, the CIS kernel proposed by Naesseth *et al.* (2020). Although not mentioned by Naesseth *et al.* (2020), this kernel has been previously proposed as the iterated sequential importance resampling (i-SIR) by Andrieu *et al.* (2018) with its corresponding geometric convergence rate as

$$
d_{\text{TV}}\left(P_\lambda^k\left(\boldsymbol{\eta}, \cdot\right), \Pi\right) = d_{\text{TV}}\left(K_\lambda^k\left(\mathbf{z}, \cdot\right), \pi\right) \leq \left(1 - \frac{N-1}{2w^* + N - 2}\right)^k.
$$

**Bound on the Gradient Variance**   The bound on the gradient variance is straightforward given Assumption 3. For simplicity, we denote the rejection state as $\mathbf{z}_t^{(1)} = \mathbf{z}_{t-1}$. Then,

$$
\begin{aligned}
&\mathbb{E}\left[\|\mathbf{g}_{t,\text{MSC}}\|^2 \,\middle|\, \mathcal{F}_{t-1}\right] \\
&= \mathbb{E}_{\mathbf{z}_t \sim K_{\boldsymbol{\lambda}_{t-1}}\left(\mathbf{z}_{t-1}, \cdot\right)}\left[\|\mathbf{s}\left(\lambda; \mathbf{z}_t\right)\|^2 \,\middle|\, \boldsymbol{\lambda}_{t-1}, \mathbf{z}_{t-1}\right] \\
&= \int \sum_{n=1}^N \frac{w\left(\mathbf{z}_t^{(n)}\right)}{\sum_{m=1}^N w\left(\mathbf{z}_t^{(m)}\right)} \left\|\mathbf{s}\left(\lambda; \mathbf{z}_t^{(n)}\right)\right\|^2 \prod_{n=2}^N q\left(d\mathbf{z}_t^{(n)}; \boldsymbol{\lambda}_{t-1}\right) \qquad \textit{Andrieu et al. (2018)} \\
&\leq L^2 \int \sum_{n=1}^N \frac{w\left(\mathbf{z}_t^{(n)}\right)}{\sum_{m=1}^N w\left(\mathbf{z}_t^{(m)}\right)} \prod_{n=2}^N q\left(d\mathbf{z}_t^{(n)}; \boldsymbol{\lambda}_{t-1}\right) \qquad \textit{Assumption 3} \\
&= L^2 \int \prod_{n=2}^N q\left(d\mathbf{z}^{(n)}; \boldsymbol{\lambda}_{t-1}\right) \\
&= L^2.
\end{aligned}
$$

□

**Lemma 3.** *Let the importance weight be defined as $w(\mathbf{z}) = \pi(\mathbf{z})/q(\mathbf{z})$. The variance of the importance weights is related to the $\chi^2$ divergence as*

$$\mathbb{V}_q w(\mathbf{z}) = d_{\chi^2}(\pi \| q).$$

*Proof.*

$$\mathbb{V}_q w(\mathbf{z}) = \mathbb{E}_q\left[\left(w(\mathbf{z}) - \mathbb{E}_q[w(\mathbf{z})]\right)^2\right] = \mathbb{E}_q\left[\left(w(\mathbf{z}) - 1\right)^2\right] = \int\left(\frac{\pi(\mathbf{z})}{q(\mathbf{z})} - 1\right)^2 q(d\mathbf{z}) = d_{\chi^2}(\pi \| q).$$

$\square$

**Theorem 2.** (Cardoso *et al.*, 2022) The gradient variance of MSC-RB is bounded as

$$\mathbb{E}\left[\|\mathbf{g}_{t,\text{MSC-RB}}\|_2^2 \mid \mathcal{F}_{t-1}\right] \le 4L^2\left[\frac{1}{N-1} d_{\chi^2}(\pi \| q(\cdot; \boldsymbol{\lambda}_{t-1})) + \mathcal{O}\left(N^{-3/2} + \gamma^{t-1}/N-1\right)\right] + \|\boldsymbol{\mu}\|_2^2,$$

where $\boldsymbol{\mu} = \mathbb{E}_\pi \mathbf{s}(\boldsymbol{\lambda}; \mathbf{z})$, $\gamma = 2w^*/(2w^* + N - 2)$ is the mixing rate of the Rao-Blackwellized CIS kernel, and $d_{\chi^2}(\pi \| q) = \int(\pi/q - 1)^2 q(d\mathbf{z})$ is the $\chi^2$ divergence.

*Proof.* Rao-Blackwellization of the CIS kernel is to reuse the importance weights $w(\mathbf{z}) = \pi(\mathbf{z})/q_{\text{def.}}(\mathbf{z})$ internally used by the kernel when forming the estimator. That is, the gradient is estimated as

$$\mathbf{z}_t^{(n)} \mid \boldsymbol{\lambda}_{t-1} \sim q(\cdot; \boldsymbol{\lambda}_{t-1})$$

$$\mathbf{g}_{t,\text{MSC-RB}} = -\sum_{n=2}^N \frac{w\left(\mathbf{z}_t^{(n)}\right)}{\sum_{m=2}^N w\left(\mathbf{z}_t^{(n)}\right) + w(\mathbf{z}_{t-1})} \mathbf{s}\left(\mathbf{z}_t^{(n)}\right) + \frac{w(\mathbf{z}_{t-1})}{\sum_{m=2}^N w\left(\mathbf{z}_t^{(n)}\right) + w(\mathbf{z}_{t-1})} \mathbf{s}(\mathbf{z}_{t-1}).$$

By Lemma 2, the second moment of the gradient is bounded as

$$\mathbb{E}\left[\|\mathbf{g}_{t,\text{MSC-RB}}\|_2^2 \mid \mathcal{F}_{t-1}\right] = \text{MSE}\left[\mathbf{g}_{t,\text{MSC-RB}} \mid \mathcal{F}_{t-1}\right] + 2\,\text{Bias}\left[\mathbf{g}_{t,\text{MSC-RB}} \mid \mathcal{F}_{t-1}\right]^\top \|\boldsymbol{\mu}\|_2 + \|\boldsymbol{\mu}\|_2^2$$

$$\le \text{MSE}\left[\mathbf{g}_{t,\text{MSC-RB}} \mid \mathcal{F}_{t-1}\right] + 2L\,\text{Bias}\left[\mathbf{g}_{t,\text{MSC-RB}} \mid \mathcal{F}_{t-1}\right] + \|\boldsymbol{\mu}\|_2^2. \quad (6)$$

Cardoso *et al.* (2022, Theorem 3) show that the mean-squared error of this estimator, which they call bias reduced self-normalized importance sampling, is bounded as

$$\text{MSE}\left[\mathbf{g}_{t,\text{MSC-RB}} \mid \mathcal{F}_{t-1}\right] \le 4L^2\left[(1 + \epsilon^2)\frac{1}{N-1}\mathbb{V}_{\mathbf{z} \sim q_{\text{def.}}(\cdot; \boldsymbol{\lambda}_{t-1})}\left[w(\mathbf{z}) \mid \boldsymbol{\lambda}_{t-1}\right]\right.$$

$$\left. + (1 + \epsilon^{-2})\frac{1}{N^2}(1 + w^*)^2\right],$$

for some arbitrary constant $\epsilon^2$. The first term is identical to the variance of an $N - 1$-sample self-normalized importance sampling estimator (Agapiou *et al.*, 2017), while the second term is the added variance due to "rejections."

Since the variance of the importance weights is well known to be related to the $\chi^2$ divergence,

$$\text{MSE}\left[\mathbf{g}_{t,\text{MSC-RB}} \mid \mathcal{F}_{t-1}\right] \le 4L^2\left[(1 + \epsilon^2)\frac{1}{N-1}d_{\chi^2}(\pi \| q(\cdot; \boldsymbol{\lambda}_{t-1}))\right.$$

$$\left. + (1 + \epsilon^{-2})\frac{1}{N^2}(1 + w^*)^2\right]. \qquad \textit{Lemma 3}$$

For $\epsilon^2$, Cardoso *et al.* choose $\epsilon^2 = (N-1)^{-1/2}$, which results in their stated bound

$$\text{MSE}\left[\mathbf{g}_{t,\text{MSC-RB}} \mid \mathcal{F}_{t-1}\right] \le 4L^2\left[\frac{1}{N-1}d_{\chi^2}(\pi \| q(\cdot; \boldsymbol{\lambda}_{t-1})) + \mathcal{O}\left(N^{-3/2}\right)\right].$$

Furthermore, they show that the bias term is bounded as

$$\text{Bias}\left[\mathbf{g}_{t,\text{MSC-RB}}\right] \le \frac{4L}{N-1}\left(d_{\chi^2}(\pi \| q(\cdot; \boldsymbol{\lambda}_{t-1})) + 1 + w^*\right)\left(\frac{2w^*}{2w^* + N - 2}\right)^{t-1}.$$

Combining both the bias and the mean-squared error to Equation (6), we obtain the bound

$$\mathbb{E}\left[\left\|\mathbf{g}_{t,\text{MSC-RB}}\right\|_2^2 \,\middle|\, \mathcal{F}_{t-1}\right]$$

$$\leq 4L^2\left[\frac{1}{N-1}d_{\chi^2}(\pi \parallel q(\cdot;\boldsymbol{\lambda}_{t-1}))\right.$$

$$\left. + \frac{1}{N-1}\left(d_{\chi^2}(\pi \parallel q(\cdot;\boldsymbol{\lambda}_{t-1})) + 1 + w^*\right)\left(\frac{2w^*}{2w^* + N - 2}\right)^{t-1} + \mathcal{O}\left(N^{-3/2}\right)\right]$$

$$= 4L^2\left[\frac{1+\gamma^{t-1}}{N-1}d_{\chi^2}(\pi \parallel q(\cdot;\boldsymbol{\lambda}_{t-1})) + \frac{\gamma^{t-1}}{N-1} + \frac{\gamma^{t-1}w^*}{N-1} + \mathcal{O}\left(N^{-3/2}\right)\right].$$

$\square$

**Lemma 4.** *For* $w^* = \sup_{\mathbf{z}} w(\mathbf{z})$, $\lambda(\cdot)$ *in Equation (3) is bounded as*

$$\max\left(1 - \frac{1}{w}, 0\right) \leq \lambda(w) \leq 1 - \frac{1}{w^*}.$$

*Proof.* The proof can be found in the proof of Theorem 3 of Smith & Tierney (1996). $\square$

**Lemma 5.** *For* $w^* = \sup_{\mathbf{z}} w(\mathbf{z})$, $T_n(\cdot)$ *in Equation (3) is bounded as*

$$T_n(w) \leq \frac{n}{w}\left(1 - \frac{1}{w^*}\right)^{n-1}.$$

*Proof.*

$$T_n(w) = \int_w^\infty \frac{n}{v^2}\lambda^{n-1}(v)\,dv \qquad \text{Equation (3)}$$

$$\leq \int_w^\infty \frac{n}{v^2}\left(1 - \frac{1}{w^*}\right)^{n-1}dv \qquad \text{Lemma 4}$$

$$= n\left(1 - \frac{1}{w^*}\right)^{n-1}\int_w^\infty \frac{1}{v^2}\,dv$$

$$= n\left(1 - \frac{1}{w^*}\right)^{n-1}\left(-\frac{1}{v}\bigg|_w^\infty\right)$$

$$= \frac{n}{w}\left(1 - \frac{1}{w^*}\right)^{n-1}.$$

$\square$

**Lemma 6.** *For a positive test function* $f : \mathcal{Z} \to \mathbb{R}^+$, *the estimate of a* $\pi$-*invariant independent Metropolis-Hastings kernel* $K$ *with a proposal* $q$ *is bounded as*

$$\mathbb{E}_{K^n(\mathbf{z},\cdot)}[f \mid \mathbf{z}] \leq n\,r^{n-1}\mathbb{E}_q[f] + r^n f(\mathbf{z}),$$

*where* $w(\mathbf{z}) = \pi(\mathbf{z})/q(\mathbf{z})$ *and* $r = 1 - 1/w^*$ *for* $w^* = \sup_{\mathbf{z}} w(\mathbf{z})$.

*Proof.*

$$\mathbb{E}_{K^n(\mathbf{z},\cdot)}[f \mid \mathbf{z}]$$

$$= \int T_n(w(\mathbf{z}) \vee w(\mathbf{z}'))\,f(\mathbf{z}')\,\pi(\mathbf{z}')\,d\mathbf{z}' + \lambda^n(w(\mathbf{z}))\,f(\mathbf{z}) \qquad \text{Equation (2)}$$

$$\leq \int \frac{n}{w(\mathbf{z}) \vee w(\mathbf{z}')}\left(1 - \frac{1}{w^*}\right)^{n-1}f(\mathbf{z}')\,\pi(\mathbf{z}')\,d\mathbf{z}' + \lambda^n(w(\mathbf{z}))\,f(\mathbf{z}) \quad \text{Lemma 5}$$

$$\leq \int \frac{n}{w(\mathbf{z}')}\left(1 - \frac{1}{w^*}\right)^{n-1}f(\mathbf{z}')\,\pi(\mathbf{z}')\,d\mathbf{z}' + \lambda^n(w(\mathbf{z}))\,f(\mathbf{z}) \qquad \frac{1}{w(\mathbf{z}) \vee w(\mathbf{z}')} \leq \frac{1}{w(\mathbf{z}')}$$

$$= n\left(1 - \frac{1}{w^*}\right)^{n-1} \int \frac{1}{w(\mathbf{z}')} f(\mathbf{z}') \pi(\mathbf{z}') d\mathbf{z}' + \lambda^n(w(\mathbf{z})) f(\mathbf{z})$$

$$= n\left(1 - \frac{1}{w^*}\right)^{n-1} \int f(\mathbf{z}') q(\mathbf{z}') d\mathbf{z}' + \lambda^n(w(\mathbf{z})) f(\mathbf{z}) \qquad \text{Definition of } w(\mathbf{z})$$

$$\leq n\left(1 - \frac{1}{w^*}\right)^{n-1} \int f(\mathbf{z}') q(\mathbf{z}') d\mathbf{z}' + \left(1 - \frac{1}{w^*}\right)^n f(\mathbf{z}) \qquad \text{Lemma 4}$$

$$= n\left(1 - \frac{1}{w^*}\right)^{n-1} \mathbb{E}_q[f] + \left(1 - \frac{1}{w^*}\right)^n f(\mathbf{z}).$$

$\square$

**Lemma 7.** *Let a $\Pi$-invariant Markov chain kernel $P$ be geometrically ergodic as*

$$d_{\mathrm{TV}}\left(P^n(\boldsymbol{\eta}_0, \cdot), \Pi\right) \leq C\rho^n.$$

*Furthermore, let $\hat{\mathbf{g}} = \mathbf{g}(\boldsymbol{\eta})$ with $\boldsymbol{\eta} \sim P^n(\boldsymbol{\eta}_0, \cdot)$ be the estimator of $\mathbb{E}_\Pi \mathbf{g}$ for some function $\mathbf{g} : \mathrm{H} \to \mathbb{R}^D$ bounded as $\|\mathbf{g}\|_2 \leq L$. The bias of $\mathbf{g}$, defined as $\mathrm{Bias}[\hat{\mathbf{g}}] = \mathbb{E}\|\hat{\mathbf{g}} - \mathbb{E}_\pi \mathbf{g}\|$, is bounded as*

$$\mathrm{Bias}[\hat{\mathbf{g}}] \leq 2\sqrt{D} L C \rho^n.$$

*Proof.*

$$\mathrm{Bias}[\hat{\mathbf{g}}] = \left\|\mathbb{E}_{P^n(\boldsymbol{\eta}_0, \cdot)} \mathbf{g} - \mathbb{E}_\Pi \mathbf{g}\right\|_2$$

$$\leq \sqrt{D}\left\|\mathbb{E}_{P^n(\boldsymbol{\eta}_0, \cdot)} \mathbf{g} - \mathbb{E}_\Pi \mathbf{g}\right\|_\infty \qquad \text{for } \mathbf{x} \in \mathbb{R}^D, \|\mathbf{x}\|_2 \leq \sqrt{D}\|\mathbf{x}\|_\infty$$

$$\leq \sqrt{D} L \sup_{|h| \leq 1} \left|\mathbb{E}_{P^n(\boldsymbol{\eta}_0, \cdot)} h - \mathbb{E}_\Pi h\right| \qquad \|\mathbf{g}\|_\infty \leq \|\mathbf{g}\|_2 \leq L$$

$$= 2\sqrt{D} L \, d_{\mathrm{TV}}\left(P^n(\boldsymbol{\eta}^{(0)}, \cdot), \pi\right) \qquad \text{Definition of } d_{\mathrm{TV}}$$

$$= 2\sqrt{D} L C \rho^n. \qquad \text{Geometric ergodicity}$$

$\square$

**Theorem 3.** JSA (Ou & Song, 2020) is obtained by defining

$$P_\lambda^k(\boldsymbol{\eta}, d\boldsymbol{\eta}') = K_\lambda^{N(k-1)+1}\left(\mathbf{z}^{(1)}, d\mathbf{z}'^{(1)}\right) K_\lambda^{N(k-1)+2}\left(\mathbf{z}^{(2)}, d\mathbf{z}'^{(2)}\right) \cdot \ldots \cdot K_\lambda^{N(k-1)+N}\left(\mathbf{z}^{(N)}, d\mathbf{z}'^{(N)}\right)$$

with $\boldsymbol{\eta}_t = \left(\mathbf{z}_t^{(1)}, \mathbf{z}_t^{(2)}, \ldots, \mathbf{z}_t^{(N)}\right)$. Then, given Assumption 2 and 3, the mixing rate and the gradient variance bounds are

$$d_{\mathrm{TV}}\left(P_\lambda^k(\boldsymbol{\eta}, \cdot), \Pi\right) \leq C(r, N)\, r^{kN} \quad \text{and} \quad \mathbb{E}\left[\|\mathbf{g}_{t,\mathrm{JSA}}\|_2^2 \,\big|\, \mathcal{F}_{t-1}\right] \leq L^2\left[\frac{1}{2} + \frac{3}{2}\frac{1}{N} + \mathcal{O}\left(1/w^* + r^{tN}\right)\right] + C_{\mathrm{cov}} + \|\boldsymbol{\mu}\|_2^2,$$

where $\boldsymbol{\mu} = \mathbb{E}_\pi \mathbf{s}(\lambda; \mathbf{z})$, $C_{\mathrm{cov}} = \frac{2}{N^2}\sum_{n=2}^N \sum_{m=1}^{n-1} \mathrm{Cov}\left(\mathbf{s}\left(\lambda; \mathbf{z}^{(n)}\right), \mathbf{s}\left(\lambda; \mathbf{z}^{(m)}\right) \big| \mathcal{F}_{t-1}\right)$ is the sum of the covariance between the samples, $w^* = \sup_{\mathbf{z}} \pi(\mathbf{z})/q_{\mathrm{def.}}(\mathbf{z}; \lambda)$, and $C(r, N) > 0$ is a finite constant.

*Proof.* JSA is described in Algorithm 5. At each iteration, it performs $N$ MCMC transitions and uses the $N$ intermediate states to estimate the gradient. That is,

$$\mathbf{z}_t^{(1)} \mid \mathbf{z}_{t-1}^{(N)}, \boldsymbol{\lambda}_{t-1} \sim K_{\boldsymbol{\lambda}_{t-1}}\left(\mathbf{z}_{t-1}^{(N)}, \cdot\right)$$

$$\mathbf{z}_t^{(2)} \mid \mathbf{z}_t^{(1)}, \boldsymbol{\lambda}_{t-1} \sim K_{\boldsymbol{\lambda}_{t-1}}\left(\mathbf{z}_t^{(1)}, \cdot\right)$$

$$\vdots$$

$$\mathbf{z}_t^{(N)} \mid \mathbf{z}_t^{(N-1)}, \boldsymbol{\lambda}_{t-1} \sim K_{\boldsymbol{\lambda}_{t-1}}\left(\mathbf{z}_t^{(N-1)}, \cdot\right)$$

$$\mathbf{g}_{t,\mathrm{JSA}} = -\frac{1}{N}\sum_{n=1}^N \mathbf{s}\left(\lambda, \mathbf{z}_t^{(n)}\right),$$

where $K_{\lambda_{t-1}}^n$ is an $n$-transition IMH kernel using $q_{\text{def.}}(\cdot; \lambda_{t-1})$. Under Assumption 2, an IMH kernel is uniformly geometrically ergodic (Mengersen & Tweedie, 1996; Wang, 2022) as

$$d_{\text{TV}}\left(K_\lambda^k(\mathbf{z}, \cdot), \pi\right) \le r^k \tag{7}$$

for any $\mathbf{z} \in \mathcal{Z}$.

**Ergodicity of the Markov Chain** The state transitions of the Markov chain samples $\mathbf{z}^{(1:N)}$ are visualized as

|  | $\mathbf{z}_t^{(1)}$ | $\mathbf{z}_t^{(2)}$ | $\mathbf{z}_t^{(3)}$ | ... | $\mathbf{z}_t^{(N)}$ |
|---|---|---|---|---|---|
| $t = 1$ | $K_{\lambda_1}\left(\mathbf{z}_0, d\mathbf{z}_1^{(1)}\right)$ | $K_{\lambda_1}^2\left(\mathbf{z}_0, d\mathbf{z}_1^{(2)}\right)$ | $K_{\lambda_1}^3\left(\mathbf{z}_0, d\mathbf{z}_1^{(3)}\right)$ | ... | $K_{\lambda_1}^N\left(\mathbf{z}_0, d\mathbf{z}_1^{(N)}\right)$ |
| $t = 2$ | $K_{\lambda_2}^{N+1}\left(\mathbf{z}_0, d\mathbf{z}_2^{(1)}\right)$ | $K_{\lambda_2}^{N+2}\left(\mathbf{z}_0, d\mathbf{z}_2^{(2)}\right)$ | $K_{\lambda_2}^{N+3}\left(\mathbf{z}_0, d\mathbf{z}_2^{(3)}\right)$ | ... | $K_{\lambda_2}^{2N}\left(\mathbf{z}_0, d\mathbf{z}_2^{(N)}\right)$ |
| $\vdots$ | | | $\vdots$ | | |
| $t = k$ | $K_{\lambda_k}^{(k-1)N+1}\left(\mathbf{z}_0, d\mathbf{z}_k^{(1)}\right)$ | $K_{\lambda_k}^{(k-1)N+2}\left(\mathbf{z}_0, d\mathbf{z}_k^{(2)}\right)$ | $K_{\lambda_k}^{(k-1)N+3}\left(\mathbf{z}_0, d\mathbf{z}_k^{(3)}\right)$ | ... | $K_{\lambda_k}^{(k-1)N+N}\left(\mathbf{z}_0, d\mathbf{z}_k^{(N)}\right)$ |

where $K_\lambda(\mathbf{z}, \cdot)$ is an IMH kernel. Therefore, the $n$-step transition kernel for the vector of the Markov-chain samples $\boldsymbol{\eta} = \mathbf{z}^{(1:N)}$ is represented as

$$P_\lambda^k\left(\boldsymbol{\eta}, d\boldsymbol{\eta}'\right) = K_\lambda^{N(k-1)+1}\left(\mathbf{z}_1, d\mathbf{z}_1'\right) K_\lambda^{N(k-1)+2}\left(\mathbf{z}_2, d\mathbf{z}_2'\right) \cdot ... \cdot K_\lambda^{N(k-1)+N}\left(\mathbf{z}_N, d\mathbf{z}_N'\right).$$

Now, the convergence in total variation $d_{\text{TV}}(\cdot, \cdot)$ can be shown to decrease geometrically as

$$
\begin{aligned}
&d_{\text{TV}}\left(P_\lambda^k\left(\boldsymbol{\eta}, \cdot\right), \Pi\right) \\
&= \sup_A \left|\Pi(A) - P^k(\boldsymbol{\eta}, A)\right| \\
&\le \sup_A \left|\int_A \pi\left(d\mathbf{z}'^{(1)}\right) \times ... \times \pi\left(d\mathbf{z}'^{(N)}\right) \right. && \textit{Definition of } d_{TV} \\
&\qquad \left. - K_\lambda^{(k-1)N+1}\left(\mathbf{z}^{(1)}, d\mathbf{z}'^{(1)}\right) \times ... \times K_\lambda^{kN}\left(\mathbf{z}^{(N)}, d\mathbf{z}'^{(N)}\right)\right| \\
&\le \sup_A \sum_{n=1}^N \left|\int_A \pi\left(d\mathbf{z}^{(n)}\right) - K_\lambda^{(k-1)N+n}\left(\mathbf{z}^{(n)}, d\mathbf{z}'^{(n)}\right)\right| && \textit{Lemma 1} \\
&= \sum_{n=1}^N d_{\text{TV}}\left(K_\lambda^{(k-1)N+n}\left(\mathbf{z}^{(n)}, \cdot\right), \pi\right) && \textit{Definition of } d_{TV} \\
&\le \sum_{n=1}^N r^{(k-1)N+n} && \textit{Equation (7)} \\
&= r^{kN} r^{-N} \frac{r - r^{N+1}}{1 - r} \\
&= \frac{r\left(1 - r^N\right)}{r^N(1 - r)}\left(r^N\right)^k.
\end{aligned}
$$

Although the constant depends on $r$ and $N$, the kernel $P$ is geometrically ergodic and converges $N$ times faster than the base kernel $K$.

**Bound on the Gradient Variance** To analyze the variance of the gradient, we require detailed information about the $n$-step marginal transition kernel, which is unavailable for most MCMC kernels. Fortunately, specifically for the IMH kernel, Smith & Tierney (1996) have shown that the $n$-step marginal IMH kernel is given as Equation (2).

Furthermore, by Lemma 2, the second moment of the gradient is bounded as

$$\mathbb{E}\left[\|\mathbf{g}_{t,\text{JSA}}\|_2^2 \,\middle|\, \mathcal{F}_{t-1}\right] = \mathbb{V}\left[\mathbf{g}_{t,\text{JSA}} \mid \mathcal{F}_{t-1}\right] + \text{Bias}\left[\mathbf{g}_{t,\text{JSA}} \mid \mathcal{F}_{t-1}\right]^2 + 2\,\text{Bias}\left[\mathbf{g}_{t,\text{JSA}} \mid \mathcal{F}_{t-1}\right]\|\boldsymbol{\mu}\|_2 + \|\boldsymbol{\mu}\|_2^2$$

$$\leq \mathbb{V}\left[\mathbf{g}_{t,\text{JSA}} \mid \mathcal{F}_{t-1}\right] + \text{Bias}\left[\mathbf{g}_{t,\text{JSA}} \mid \mathcal{F}_{t-1}\right]^2 + 2L\,\text{Bias}\left[\mathbf{g}_{t,\text{JSA}} \mid \mathcal{F}_{t-1}\right] + \|\boldsymbol{\mu}\|_2^2,$$

where $\boldsymbol{\mu} = \mathbb{E}_\pi \mathbf{s}(\lambda; \mathbf{z})$. As shown in Lemma 7, the bias terms decreases in a rate of $r^{tN}$. Therefore,

$$\mathbb{E}\left[\|\mathbf{g}_{t,\text{JSA}}\|_2^2 \mid \mathcal{F}_{t-1}\right] \leq \mathbb{V}\left[\mathbf{g}_{t,\text{JSA}} \mid \mathcal{F}_{t-1}\right] + \|\boldsymbol{\mu}\|_2^2 + \mathcal{O}\left(r^{tN}\right). \quad \textit{Lemma 7}$$

Note that it is possible to obtain a tighter bound on the bias terms such that $\mathcal{O}\left(r^{tN}/N\right)$, if we directly use $(K, \mathbf{z})$ to bound the bias instead of the higher-level $(P, \boldsymbol{\eta})$ abstraction. The extra looseness comes from the use of Lemma 1.

For the variance term, we show that

$$\mathbb{V}\left[\mathbf{g}_{t,\text{JSA}} \mid \mathcal{F}_{t-1}\right]$$

$$= \mathbb{V}\left[\frac{1}{N}\sum_{n=1}^{N} \mathbf{s}\left(\lambda; \mathbf{z}_t^{(n)}\right) \bigg| \mathcal{F}_{t-1}\right]$$

$$= \frac{1}{N^2}\sum_{n=1}^{N} \mathbb{V}\left[\mathbf{s}\left(\lambda; \mathbf{z}_t^{(n)}\right) \big| \mathcal{F}_{t-1}\right] + \frac{2}{N^2}\sum_{n=2}^{N}\sum_{m=1}^{n-1} \text{Cov}\left(\mathbf{s}\left(\lambda; \mathbf{z}_t^{(n)}\right), \mathbf{s}\left(\lambda; \mathbf{z}_t^{(m)}\right) \big| \mathcal{F}_{t-1}\right)$$

$$\leq \frac{1}{N^2}\sum_{n=1}^{N} \mathbb{E}\left[\left\|\mathbf{s}\left(\lambda; \mathbf{z}_t^{(n)}\right)\right\|_2^2 \bigg| \mathcal{F}_{t-1}\right] + C_{\text{cov}}$$

$$= \frac{1}{N^2}\sum_{n=1}^{N} \mathbb{E}\left[\left\|\mathbf{s}\left(\lambda; \mathbf{z}_t^{(n)}\right)\right\|_2^2 \bigg| \mathbf{z}_{t-1}^{(N)}, \lambda_{t-1}\right] + C_{\text{cov}}$$

$$= \frac{1}{N^2}\sum_{n=1}^{N} \mathbb{E}_{\mathbf{z}_t^{(n)} \sim K_{\lambda_{t-1}}^n(\mathbf{z}_{t-1}, \cdot)}\left[\left\|\mathbf{s}\left(\lambda; \mathbf{z}_t^{(n)}\right)\right\|_2^2 \bigg| \mathbf{z}_{t-1}^{(N)}, \lambda_{t-1}\right] + C_{\text{cov}}$$

$$\leq \frac{1}{N^2}\sum_{n=1}^{N} \left[n\,r^{n-1}\mathbb{E}_{\mathbf{z} \sim q_{\text{def.}}(\cdot;\lambda_{t-1})}\left[\|\mathbf{s}(\lambda; \mathbf{z})\|_2^2\right] + r^n\left\|\mathbf{s}\left(\lambda; \mathbf{z}_{t-1}^{(N)}\right)\right\|_2^2\right] + C_{\text{cov}} \qquad \textit{Lemma 6}$$

$$\leq \frac{1}{N^2}\sum_{n=1}^{N} \left[n\,r^{n-1}L^2 + r^n L^2\right] + C_{\text{cov}} \qquad \textit{Assumption 3}$$

$$= \frac{L^2}{N^2}\sum_{n=1}^{N} \left[n\left(1 - \frac{1}{w^*}\right)^{n-1} + \left(1 - \frac{1}{w^*}\right)^n\right] + C_{\text{cov}}$$

$$= \frac{L^2}{N^2}\left[(w^*)^2 + w^* - \left(1 - \frac{1}{w^*}\right)^N\left((w^*)^2 + w^* + Nw^*\right)\right] + C_{\text{cov}}$$

$$= \frac{L^2}{N^2}\left[\frac{1}{2}N^2 + \frac{3}{2}N + \mathcal{O}\left(1/w^*\right)\right] + C_{\text{cov}} \qquad \textit{Laurent series expansion at } w^* \to \infty$$

$$= L^2\left[\frac{1}{2} + \frac{3}{2}\frac{1}{N} + \mathcal{O}\left(1/w^*\right)\right] + C_{\text{cov}},$$

where

$$C_{\text{cov}} = \frac{2}{N^2}\sum_{n=2}^{N}\sum_{m=1}^{n-1} \text{Cov}\left(\mathbf{s}\left(\lambda; \mathbf{z}_t^{(n)}\right), \mathbf{s}\left(\lambda; \mathbf{z}_t^{(m)}\right) \big| \mathbf{z}_{t-1}^{(N)}, \lambda_{t-1}\right).$$

The Laurent approximation becomes exact as $w^* \to \infty$, which is useful considering Proposition 2. $\qquad\square$

**Theorem 4.** pMCSA, our proposed scheme, is obtained by setting

$$P_\lambda^k(\boldsymbol{\eta}, d\boldsymbol{\eta}') = K_\lambda^k\left(\mathbf{z}^{(1)}, d\mathbf{z}'^{(1)}\right) K_\lambda^k\left(\mathbf{z}^{(2)}, d\mathbf{z}'^{(2)}\right) \cdot \ldots \cdot K_\lambda^k\left(\mathbf{z}^{(N)}, d\mathbf{z}'^{(N)}\right)$$

with $\boldsymbol{\eta} = \left(\mathbf{z}^{(1)}, \mathbf{z}^{(2)}, \ldots, \mathbf{z}^{(N)}\right)$. Then, given Assumption 2 and 3, the mixing rate and the gradient variance bounds are

$$d_{\text{TV}}\left(P_\lambda^k(\boldsymbol{\eta}, \cdot), \Pi\right) \leq C(N)\,r^k \quad \text{and} \quad \mathbb{E}\left[\|\mathbf{g}_{t,\text{pMCSA}}\|_2^2 \mid \mathcal{F}_{t-1}\right] \leq L^2\left[\frac{1}{N} + \frac{1}{N}\left(1 - \frac{1}{w^*}\right)\right] + \mathcal{O}\left(r^t\right) + \|\boldsymbol{\mu}\|_2^2,$$

where $\boldsymbol{\mu} = \mathbb{E}_\pi \mathbf{s}(\lambda; \mathbf{z})$, $w^* = \sup_{\mathbf{z}} \pi(\mathbf{z})/q_{\text{def.}}(\mathbf{z}; \lambda)$ and $C(N) > 0$ is a finite constant.

*Proof.* Our proposed scheme, pMCSA, is described in Algorithm 5. At each iteration, our scheme performs a single MCMC transition for each of the $N$ samples, or chains, to estimate the gradient. That is,

$$\mathbf{z}_t^{(1)} \mid \mathbf{z}_{t-1}^{(1)}, \boldsymbol{\lambda}_{t-1} \sim K_{\boldsymbol{\lambda}_{t-1}}\left(\mathbf{z}_{t-1}^{(1)}, \cdot\right)$$

$$\mathbf{z}_t^{(2)} \mid \mathbf{z}_{t-1}^{(2)}, \boldsymbol{\lambda}_{t-1} \sim K_{\boldsymbol{\lambda}_{t-1}}\left(\mathbf{z}_{t-1}^{(2)}, \cdot\right)$$

$$\vdots$$

$$\mathbf{z}_t^{(N)} \mid \mathbf{z}_{t-1}^{(N)}, \boldsymbol{\lambda}_{t-1} \sim K_{\boldsymbol{\lambda}_{t-1}}\left(\mathbf{z}_{t-1}^{(N)}, \cdot\right)$$

$$\mathbf{g}_{t,\mathrm{pMCSA}} = -\frac{1}{N} \sum_{n=1}^{N} \mathbf{s}\left(\lambda, \mathbf{z}_t^{(n)}\right),$$

where $K_{\boldsymbol{\lambda}_{t-1}}^n$ is an $n$-transition IMH kernel using $q_{\mathrm{def.}}\left(\cdot; \boldsymbol{\lambda}_{t-1}\right)$.

**Ergodicity of the Markov Chain**  Since our kernel operates the same MCMC kernel $K_\lambda$ for each of the $N$ parallel Markov chains, the $n$-step marginal kernel $P_\lambda$ can be represented as

$$P_\lambda^k\left(\boldsymbol{\eta}, d\boldsymbol{\eta}'\right) = K_\lambda^k\left(\mathbf{z}^{(1)}, d\mathbf{z}'^{(1)}\right) K_\lambda^k\left(\mathbf{z}^{(2)}, d\mathbf{z}'^{(2)}\right) \cdot \ldots \cdot K_\lambda^k\left(\mathbf{z}^{(N)}, d\mathbf{z}'^{(N)}\right).$$

Then, the convergence in total variation $d_{\mathrm{TV}}\left(\cdot, \cdot\right)$ can be shown to decrease geometrically as

$$
\begin{aligned}
& d_{\mathrm{TV}}\left(K_\lambda^k\left(\boldsymbol{\eta}, \cdot\right), \Pi\right) \\
&= \sup_A \left| \Pi\left(A\right) - P_\lambda^k\left(\boldsymbol{\eta}, A\right) \right| && \textit{Definition of } d_{TV} \\
&\leq \sup_A \Big| \int_A \pi\left(d\mathbf{z}_1'\right) \cdot \ldots \cdot \pi\left(d\mathbf{z}_N'\right) \\
& \qquad\quad - K_\lambda^k\left(\mathbf{z}_1, d\mathbf{z}_1'\right) \cdot \ldots \cdot K_\lambda^k\left(\mathbf{z}_N, d\mathbf{z}_N'\right) \Big| \\
&\leq \sup_A \sum_{n=1}^{N} \left| \int_A \pi\left(d\mathbf{z}_k'\right) - K_\lambda^k\left(\mathbf{z}_n, d\mathbf{z}_n'\right) \right| && \textit{Lemma 1} \\
&= \sum_{n=1}^{N} d_{\mathrm{TV}}\left(K_\lambda^k\left(\mathbf{z}_n, \cdot\right), \pi\right) && \textit{Equation (7)} \\
&\leq \sum_{n=1}^{N} r^k && \textit{Geometric ergodicity} \\
&= N\, r^k.
\end{aligned}
$$

**Bound on the Gradient Variance**  By Lemma 2, the second moment of the gradient is bounded as

$$\mathbb{E}\left[\left\|\mathbf{g}_{t,\mathrm{pMCSA}}\right\|_2^2 \mid \mathcal{F}_{t-1}\right] = \mathbb{V}\left[\mathbf{g}_{t,\mathrm{pMCSA}} \mid \mathcal{F}_{t-1}\right] + \mathrm{Bias}\left[\mathbf{g}_{t,\mathrm{pMCSA}} \mid \mathcal{F}_{t-1}\right]^2 + 2\,\mathrm{Bias}\left[\mathbf{g}_{t,\mathrm{pMCSA}} \mid \mathcal{F}_{t-1}\right]\left\|\boldsymbol{\mu}\right\|_2 + \left\|\boldsymbol{\mu}\right\|_2^2$$

$$\leq \mathbb{V}\left[\mathbf{g}_{t,\mathrm{pMCSA}} \mid \mathcal{F}_{t-1}\right] + \mathrm{Bias}\left[\mathbf{g}_{t,\mathrm{pMCSA}} \mid \mathcal{F}_{t-1}\right]^2 + 2\,L\,\mathrm{Bias}\left[\mathbf{g}_{t,\mathrm{pMCSA}} \mid \mathcal{F}_{t-1}\right] + \left\|\boldsymbol{\mu}\right\|_2^2,$$

where $\boldsymbol{\mu} = \mathbb{E}_\pi \mathbf{s}\left(\lambda; \mathbf{z}\right)$. As shown in Lemma 7, the bias terms decreases in a rate of $r^t$. Therefore,

$$\mathbb{E}\left[\left\|\mathbf{g}_{t,\mathrm{pMCSA}}\right\|_2^2 \mid \mathcal{F}_{t-1}\right] \leq \mathbb{V}\left[\mathbf{g}_{t,\mathrm{pMCSA}} \mid \mathcal{F}_{t-1}\right] + \left\|\boldsymbol{\mu}\right\|_2^2 + \mathcal{O}\left(r^t\right). \quad \textit{Lemma 7}$$

As noted in the proof of Theorem 3, it is possible to obtain a tighter bound on the bias terms such that $\mathcal{O}\left(r^t/N\right)$.

The variance term is bounded as

$$
\begin{aligned}
& \mathbb{V}\left[\mathbf{g}_{t,\mathrm{pMCSA}} \mid \mathcal{F}_{t-1}\right] \\
&= \mathbb{V}\left[\frac{1}{N} \sum_{n=1}^{N} \mathbf{s}\left(\lambda; \mathbf{z}_t^{(n)}\right) \,\middle|\, \mathbf{z}_{t-1}^{(1:N)}, \boldsymbol{\lambda}_{t-1}\right]
\end{aligned}
$$

$$= \frac{1}{N^2} \sum_{n=1}^{N} \mathbb{V}\left[ \mathbf{s}\left(\lambda; \mathbf{z}_t^{(n)}\right) \middle| \mathbf{z}_{t-1}^{(1:N)}, \boldsymbol{\lambda}_{t-1} \right] \qquad\qquad \mathbf{z}_t^{(i)} \perp \mathbf{z}_t^{(j)} \; for \; i \neq j$$

$$\leq \frac{1}{N^2} \sum_{n=1}^{N} \mathbb{E}\left[ \left\| \mathbf{s}\left(\lambda; \mathbf{z}_t^{(n)}\right) \right\|_2^2 \middle| \mathbf{z}_{t-1}^{(1:N)}, \boldsymbol{\lambda}_{t-1} \right]_2$$

$$= \frac{1}{N^2} \sum_{n=1}^{N} \mathbb{E}_{\mathbf{z}_t^{(n)} \sim K_{\boldsymbol{\lambda}_{t-1}}\left(\mathbf{z}_{t-1}^{(n)}, \cdot\right)}\left[ \left\| \mathbf{s}\left(\lambda; \mathbf{z}_t^{(n)}\right) \right\|_2^2 \middle| \mathbf{z}_{t-1}^{(1:N)}, \boldsymbol{\lambda}_{t-1} \right]$$

$$\leq \frac{1}{N^2} \sum_{n=1}^{N} \left[ \mathbb{E}_{\mathbf{z} \sim q_{\mathrm{def.}}(\cdot; \boldsymbol{\lambda}_{t-1})}\left[ \left\| \mathbf{s}\left(\lambda; \mathbf{z}\right) \right\|_2^2 \right] + r \left\| \mathbf{s}\left(\lambda; \mathbf{z}_{t-1}^{(n)}\right) \right\|_2^2 \right] \qquad \textit{Lemma 6}$$

$$\leq \frac{1}{N^2} \sum_{n=1}^{N} \left[ L^2 + r\, L^2 \right] \qquad\qquad\qquad \textit{Assumption 3}$$

$$= \frac{L^2}{N^2} \sum_{n=1}^{N} \left[ 1 + r \right]$$

$$= L^2 \left[ \frac{1}{N} + \frac{1}{N} \left( 1 - \frac{1}{w^*} \right) \right].$$

$\square$

# E  Additional Experimental Results

## E.1  Bayesian Neural Network Regression

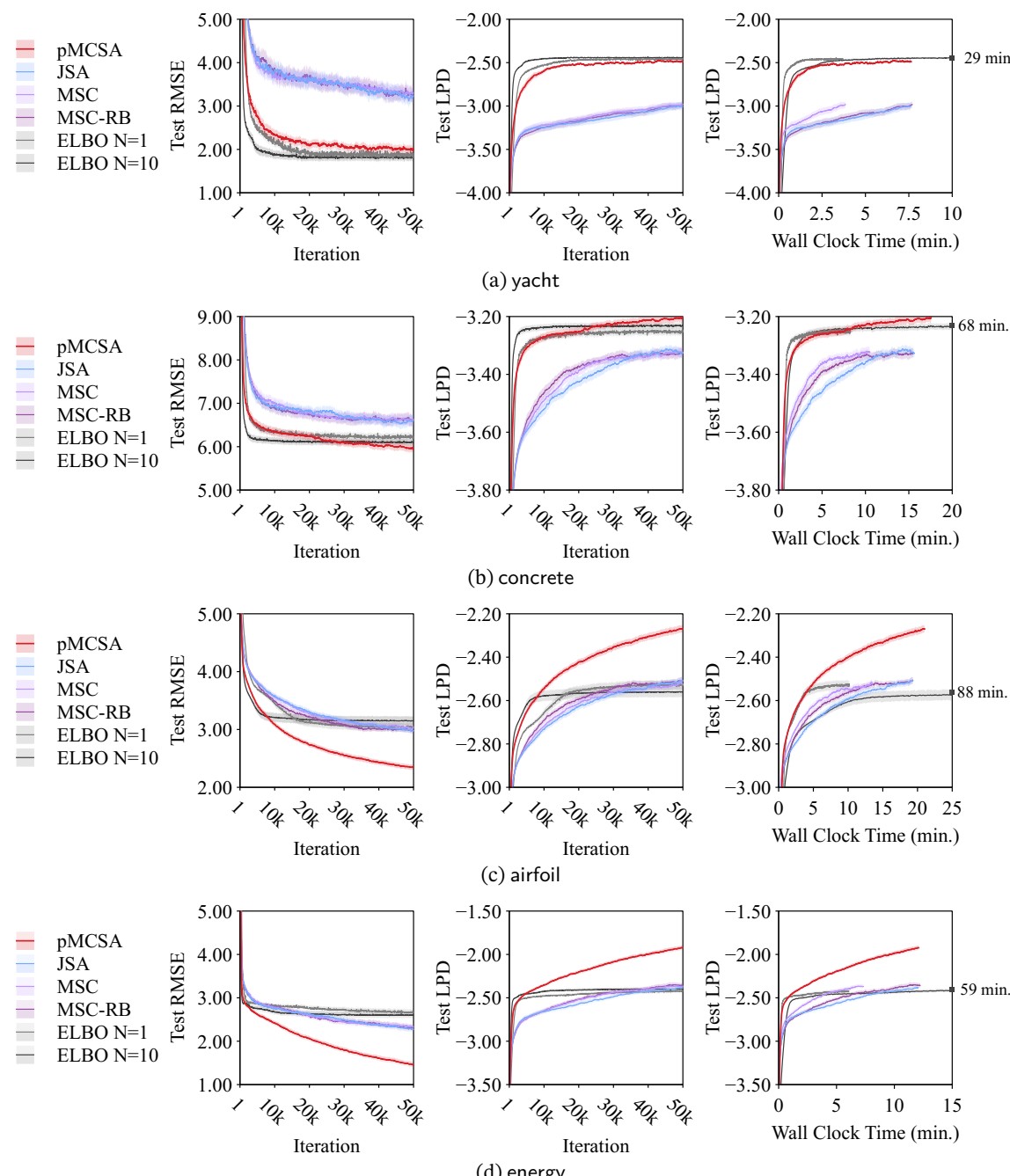

Figure 4: **Test root-mean-square error (RMSE) and test log predictive density (LPD) on Bayesian neural network regression.** The grey squares (■) mark the performance of ELBO $N = 10$ at the wall clock time shown next to it. The error bands show the 95% bootstrap confidence intervals obtained from 20 independent 90% train-test splits.

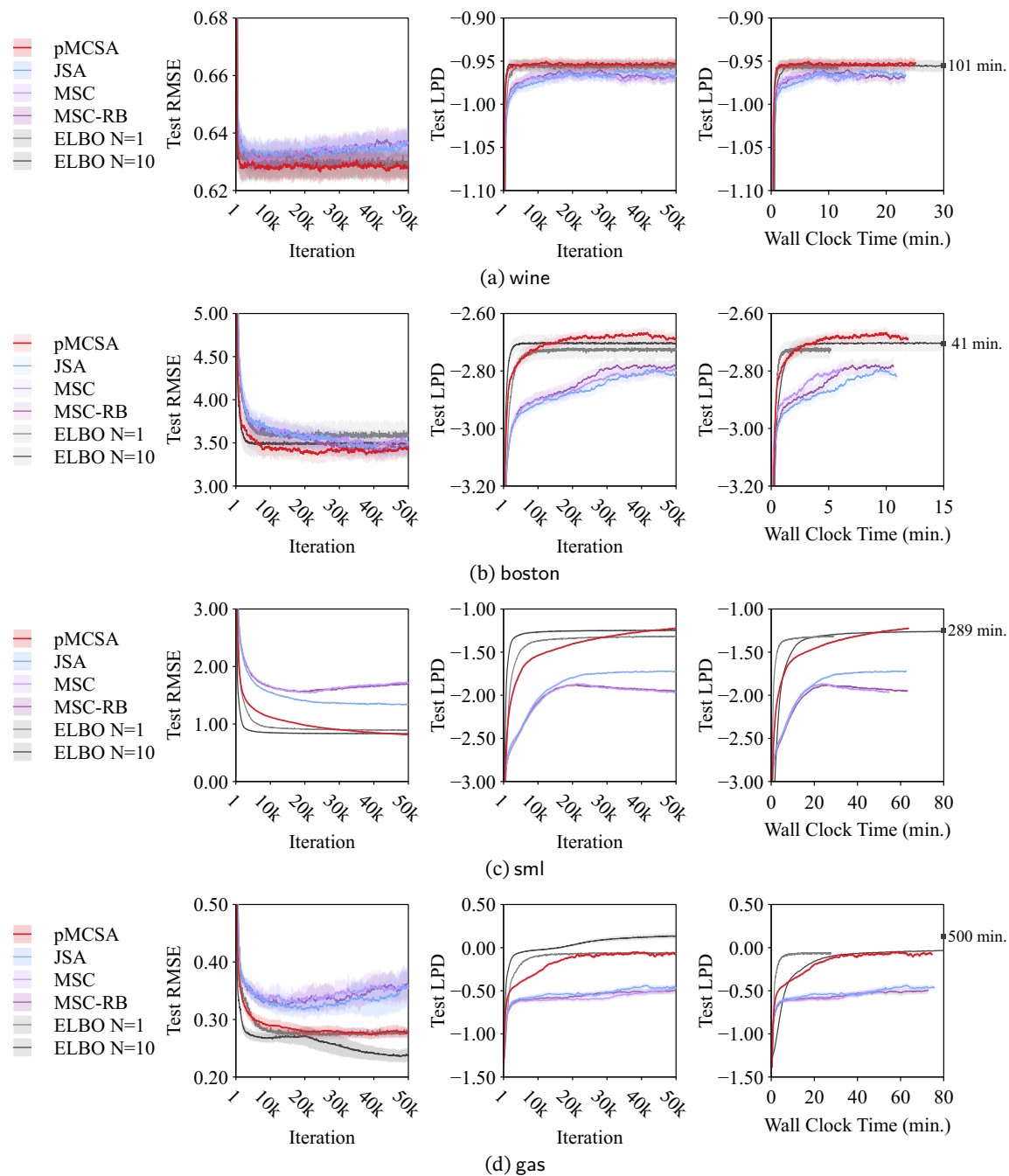

Figure 5: **(continued) Test root-mean-square error (RMSE) and test log predictive density (LPD) on Bayesian neural network regression.** The grey squares (■) mark the performance of ELBO $N = 10$ at the wall clock time shown next to it. The error bands show the 95% bootstrap confidence intervals obtained from 20 independent 90% train-test splits.

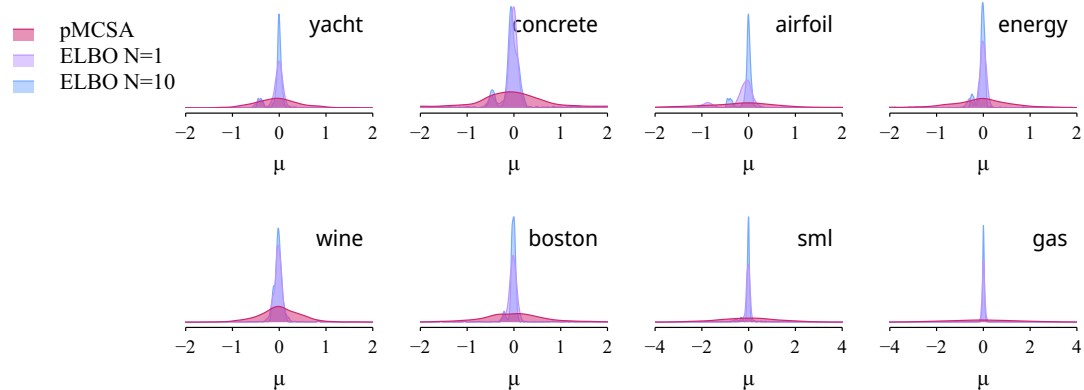

Figure 6: **Distribution of the variational posterior mean of the BNN weights.** The density was estimated with a Gaussian kernel and the bandwidth was selected with Silverman's rule

## E.2 Robust Gaussian Process Regression

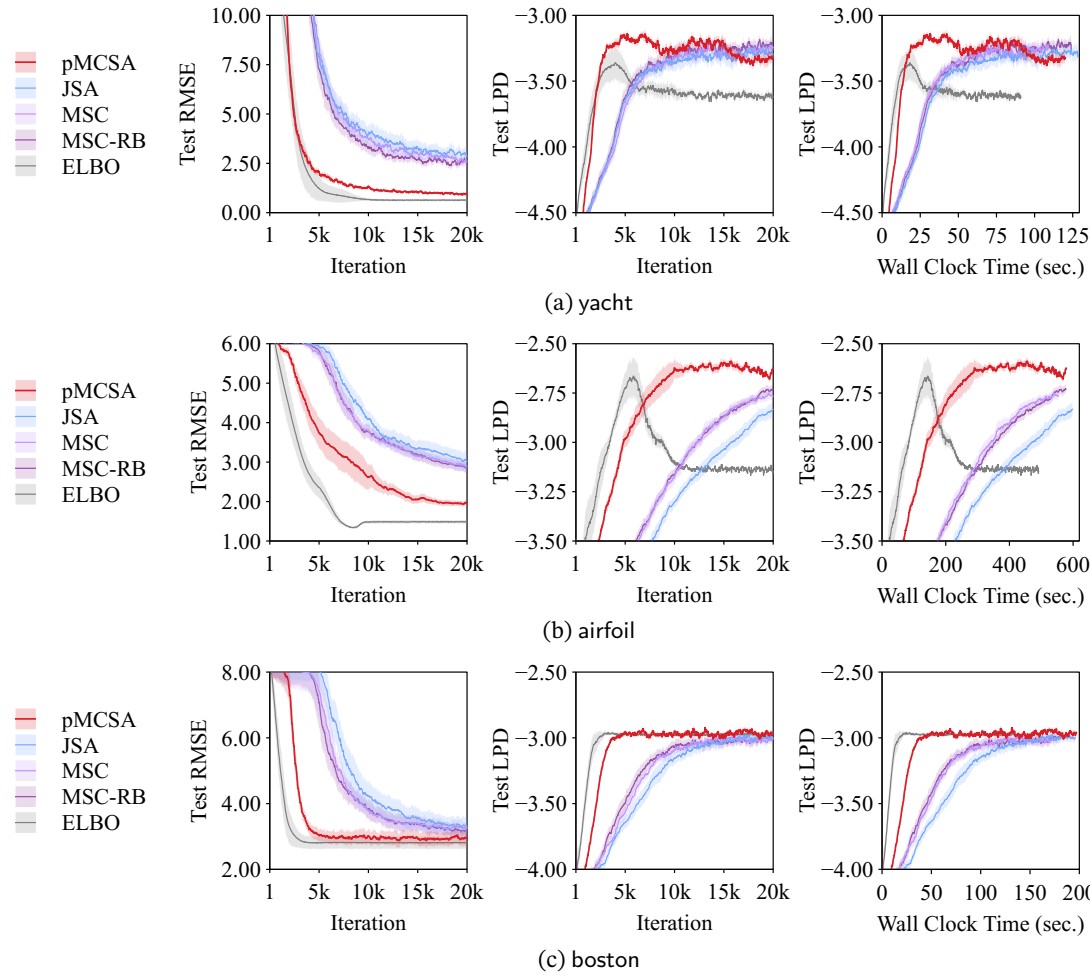

Figure 7: **Test root-mean-square error (RMSE) and test log predictive density (LPD) on robust Gaussian process regression.** The error bands shows the 95% bootstrap confidence interval obtained from 20 repetitions.

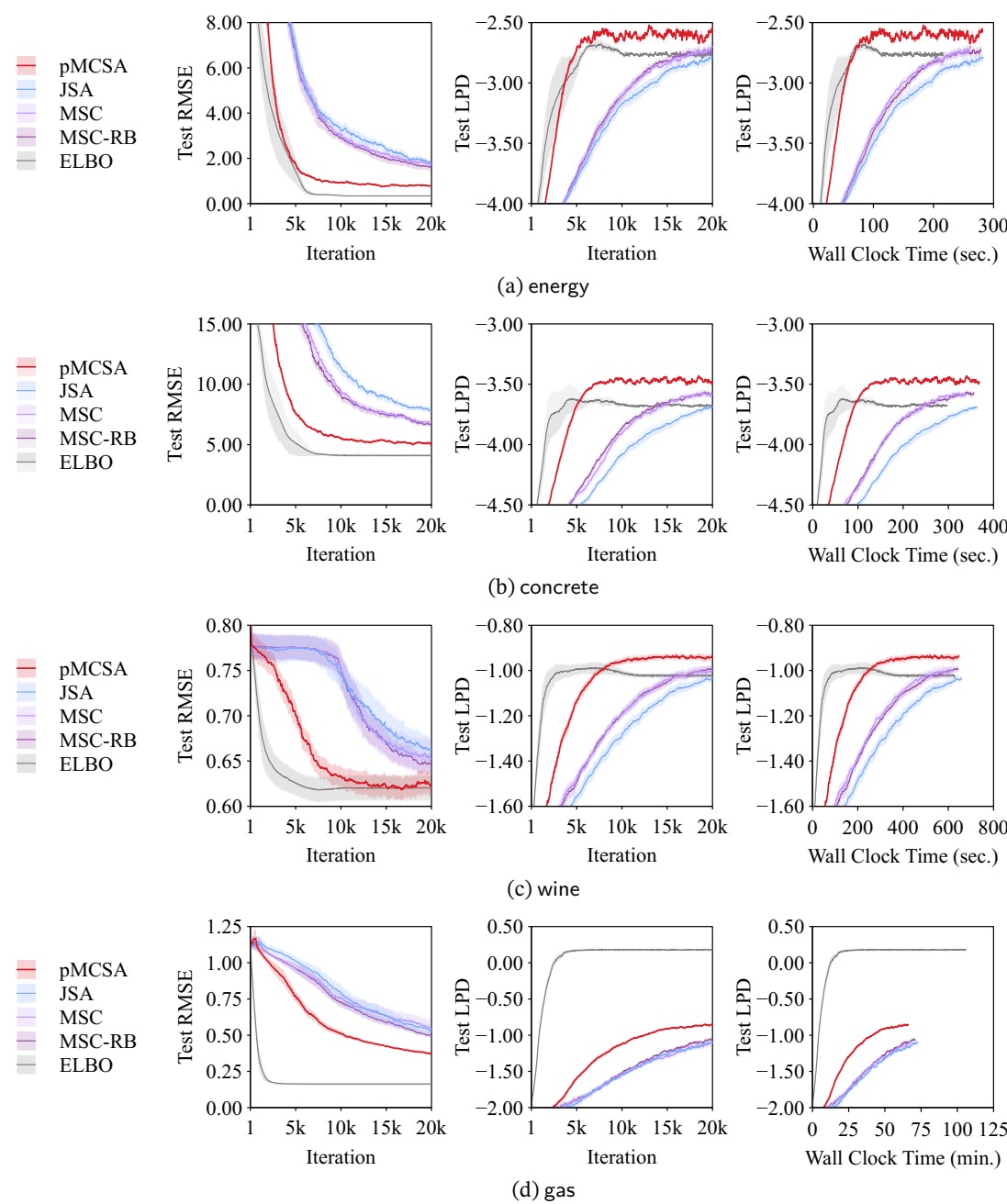

Figure 8: **(continued) Test root-mean-square error (RMSE) and test log predictive density (LPD) on robust Gaussian process regression.** The error bands show the 95% bootstrap confidence intervals obtained from 20 independent 90% train-test splits.