# OpenReview forum: "Markov Chain Score Ascent: A Unifying Framework of Variational Inference with Markovian Gradients"
_NeurIPS.cc/2022/Conference — NeurIPS 2022 Accept_

### Official Review · Reviewer_ceLV · 2022-06-27

**Rating:** 7
**Confidence:** 4
**Soundness:** 4 excellent
**Presentation:** 4 excellent
**Contribution:** 4 excellent

**Summary:**

The paper examines variational inference with the inclusive KL, KL(/pi, q), where required samples from /pi are obtained using MCMC.
In particular, the authors provide theoretical analysis of non-asymptotic convergence of two published methods, and establish mixing rates and bounds on gradient variance.
This is done by casting those methods as instances of Markov Chain Gradient Descent, and applying and adapting existing theoretical results.
Further based on their findings, the authors propose a novel scheme that trades off mixing rate with for reduced variance, and through that performs favorably.
A number of experiments on simple real-world data confirm the theoretical findings.


**Questions:**

One question that is not really discussed in the paper is: "Is it worth giving up the scalability of the standard variational inference for the improved performance of the discussed methods?". I think since all papers in the MCSA space lack large-scale experiments, this really remains to be seen, but I'd like to hear the authors comment on this.

289 "We consider ELBO with only 𝑁 = 1 since differentiating through the likelihood makes its per-iteration cost comparable to MCSA methods with 𝑁 = 10." Should be elaborated as it is not clear where this number comes from.

294 "Our encouraging regression results suggest that incorporating methods such as inducing points (Snelson & Ghahramani, 2005) into MCSA may lead to an important new class of GP models." not clear what that means and what the basis of this is.

**Limitations:**

One problem with the results are the strong assumptions necessary (the authors do state this), however, it seems very hard to derive comparable results without any assumptions, and the experiments validate some of the findings.
In total, the paper appears to be honest about strengths and weaknesses of the analysis method.

**Strengths And Weaknesses:**

This is a very nice paper, well written, tackling a worthwhile problem, and executed with a very high level of sophistication.

The paper begins by casting two existing inclusive KL score climbing methods under the framework of Markov Chain Gradient Descent, a method to optimize an expectation of a function over an intractable distribution that works via using MCMC to draw samples from the distribution.
This is done via Proposition 1, which is a relatively simple re-writing exercise of the involved quantities.
Despite being simple, this connection allows the authors to apply and adapt existing theoretical results to establish results on mixing and gradient variance (Thm1/2).
The authors discuss the implications, and in particular the downsides of the both analysed methods.


Originality:
The authors connect existing works to the established field of Markov Chain Gradient Descent, a method to optimize an expectation of a function over an intractable distribution that works via using MCMC to draw samples from the distribution.
This is done via Proposition 1, which is a relatively simple re-writing exercise of the involved quantities.
Despite being simple, this highly original re-casting allows the authors to apply and adapt existing theoretical results to establish results on mixing and gradient variance (Thm1/2).
On top of this, the authors take their findings and use them to motivate an original method that overcomes shortcomings of the analysed methods.


Quality:
The whole paper is carried out at a very high standard.
All claims are supported by proofs and their implications are discussed in the main text.
I appreciate the discussions of the theoretical results in plain text.
It is particularly nice that the analysis of the existing work leads to a new algorithm that performs favorably in practice -- it is very nice to have theoretical results motivate new methodology.
For the new method, the authors provide a comparison of computation costs -- though it would have been nice to see this confirmed in the experiments, by e.g. plotting performance as a function of wallclock time instead of iterations.
It is also great that the authors compare various optimizers and step-sizes.


Clarity:
The paper is very well written and organized.
The proof contains justifications for most steps, which makes it very nice to read the derivations.


Significance:
The results are significant from a theoretical / unification perspective, as well is the proposed algorithm from a practical perspective (due to its strong performance).
The only real critique I have for the paper is the lack of modern large-scale experiments.
That is, deep generative models with complex architectures, e.g. on images, 3d scenes or scientific domains, and how MCSA perform here in general.
The experiments considered are pretty much all more or less solved problems with only marginal gains to be made.
A strong large-scale results would significantly improve the impact of this paper.

114 missing s in iterations
eq under 162, should say that L is score bound
line 284, verb missing ("use"?)
171 alg 4 is in appendix

---

> ### Author Response · Authors · 2022-08-01
> **Wall-clock Time Plot and Future Directions Towards Scalability**
>
> Thank you for your feedback.
>
> > though it would have been nice to see this confirmed in the experiments, by e.g. plotting performance as a function of wallclock time instead of iterations. It is also great that the authors compare various optimizers and step-sizes.
>
> Currently, experiments are plotted against wall clock time in Appendix E. We will reference these experiments more prominently in the main text. In terms of practical scalability beyond per-iteration timing, we also consider stochastic extensions of our methods to be interesting potential future work.
>
> > 289 "We consider ELBO with only 𝑁 = 1 since differentiating through the likelihood makes its per-iteration cost comparable to MCSA methods with 𝑁 = 10." Should be elaborated as it is not clear where this number comes from.
>
> We point toward the wall-clock time plots in Appendix E, where the methods can be seen to have similar computational costs. We will state this more clearly in the future version.
>
> > One question that is not really discussed in the paper is: "Is it worth giving up the scalability of the standard variational inference for the improved performance of the discussed methods?". I think since all papers in the MCSA space lack large-scale experiments, this really remains to be seen, but I'd like to hear the authors comment on this.
>
> This is indeed a very important question. Since previously proposed scalable MCMC algorithms have not been considered as a component of VI as in our work, we believe it would be worth re-evaluating them in the MCSA setting. Given the volume of past research on the subject, the list of potentially interesting candidates is pretty long, which we plan to investigate in the future. Our current paper should act as a theoretical foundation in this direction.
>
> > 294 "Our encouraging regression results suggest that incorporating methods such as inducing points (Snelson & Ghahramani, 2005) into MCSA may lead to an important new class of GP models." not clear what that means and what the basis of this is.
>
> We will clarify this in the future version’s main text. Currently, most scalable GP methods rely on sparse approximations with inducing points. We believe that combining inducing points with MCSA will be beneficial and lead to new forms of sparse GP approximations.

---

### Official Review · Reviewer_mF4s · 2022-07-11

**Rating:** 6
**Confidence:** 2
**Soundness:** 3 good
**Presentation:** 3 good
**Contribution:** 3 good

**Summary:**

This paper provides a non-asymptotic convergence analysis of methods that minimize the inclusive Kullback-Leibler (KL) divergence with stochastic gradient descent (SGD). The authors prove that these Markov chain score ascent methods are special cases of a unified framework. Based on this framework, a new scheme is proposed.

**Questions:**

None.

**Limitations:**

Yes, the authors have addressed the limitations and potential negative societal impact of their work.

**Strengths And Weaknesses:**

Unfortunately, I am somewhat of an outsider to the field of this paper (cannot check the correctness of the main part such as the non-asymptotic convergence of the proposed framework) and I feel I cannot accurately assess the importance of the proposed framework and the novel scheme. However, this paper is definitely well-organized and technically solid.

---

> ### Author Response · Authors · 2022-08-01
> **Thank you**
>
> We thank you for your service.

---

### Official Review · Reviewer_seMc · 2022-07-11

**Rating:** 6
**Confidence:** 3
**Soundness:** 3 good
**Presentation:** 3 good
**Contribution:** 2 fair

**Summary:**

This paper presents a unified framework of Markov Chain Score Ascent (MCSA) methods and a theoretical study of MCSA gradient variance for minimizing the forward KL divergence in variational inference. Based on the theoretical analysis, the paper proposes a simple modification to use parallel Markov chains, named parallel MCSA (pMCSA). Experiment results show that pMCSA achieves better empirical performance for a number of Bayesian inference problems.

**Questions:**

- Are there scenarios where the mixing rate is important and JSA could attain better performance?
- Does pMCSA also trade bias for variance during early training iterations?
- It would be beneficial to also plot MSC-RB in Figure 1, 2 to match the computational costs between different methods.

**Limitations:**

The authors adequately addressed the limitations of their work. The authors did not discuss any potential negative societal impacts of the work, but as the work is mostly theoretical it is unlikely to have immediate negative societal impacts.

**Strengths And Weaknesses:**

Strengths:
- The paper unifies previous works on forward KL minimization under the same framework.
- The study on the improvements of gradient variance w.r.t. $N$ for prior methods is useful and insightful.
- The proposed pMCSA algorithm overall achieves better empirical performance than JSA, MSC and MSC-RB.

Weaknesses:
- Analysis is largely based on the more general Markov chain gradient descent methodology, of which MCSA is only a special case. This limits the novelty and scope of the work.
- The proposed modification is relatively simplistic and similar to other parallel MCMC algorithms. Switching the kernel to MH in MSC and increasing sample size also effectively yields the same algorithm.

---

> ### Author Response · Authors · 2022-08-01
> **Response to Questions**
>
> Thank you for your feedback – responses to your questions are below.
>
> > It would be beneficial to also plot MSC-RB in Figure 1, 2 to match the computational costs between different methods.
>
> Thank you for the suggestion. We have added experiments with MSC-RB in Figure 1, 2 In our **uploaded revision**.
>
> > Does pMCSA also trade bias for variance during early training iterations?
>
> The trade-off is a “fixed” feature and kicks in regardless of the iteration as implied by Theorem 3.
>
> > Are there scenarios where the mixing rate is important and JSA could attain better performance?
>
> Yes, bias suddenly becomes important for parameterized posteriors such as VAEs since the target distribution also moves around. Although we agree that this is an important setting that many would be interested to learn about, we believe this is a separate topic worth its own article since the MCGD framework cannot be directly applied.

---

> > ### Comment · Reviewer_seMc · 2022-08-07
> > **Thanks for addressing the questions - score increase from 5 to 6**
> >
> > Thank you for addressing my questions and suggestions. These answers are helpful and addressed my concerns about mixing rate. The updated Figures 1, 2 are also more informative and highlight the benefits of the proposed approach. Therefore I increased my score from 5 to 6.

---

### Official Review · Reviewer_PECh · 2022-07-13

**Rating:** 7
**Confidence:** 3
**Soundness:** 3 good
**Presentation:** 3 good
**Contribution:** 3 good

**Summary:**

The authors provide non-asymptotic convergence rates of Markov Chain Score Ascent (MCSA) methods by casting them as instances of Markovian Chain Gradient Descent for which such rates have been recently established. Based on these results the authors hypothesize that the convergence of MCSA depends more heavily on the gradient variance than on the mixing rate of the Markov Chain and propose a parallel MCSA (pMCSA) scheme which leverages these insights. In contrast to JSA which performs $N$ transition on a single chain, pMCSA performs a single transition on $N$ independent Markov chains, hence lowering the variance of the gradient in exchange for a slower mixing rate.


**Questions:**

- Based on the analysis JSA and MSC are both heavily influenced by model misspecification (of the variational approximation) via $w^*$ while pMCSA is independent of $w^*$. Is this effect expected to vanish with increasing capacity of the variational distribution (with reasonable initialization) after some initial training time?

Regarding the convergence rates (mentioned above):
- I was specifically wondering about the $log T$ term in the numerator of the rate for Mirror Descent which I was not able to match to the rate stated in the original paper.
- I was also confused by the statement that the influence of mixing rate on the convergence rate decreases as $O(1/T^2)$ even though the stated convergence rate does depend on the mixing rate at all. I might be missing something here and would be grateful for clarification.

Some minor thing I noticed:
- Equation after line 97. Product sign between $\pi(z^{(1)}$ and $\pi(z^{(2)}$ is missing
- The gradient bound $G$ is mentioned on Line 114 but has not yet been introduced
- Line 165L MSA -> MSC


**Limitations:**

The authors clearly state their assumptions and limitations.

**Strengths And Weaknesses:**

**Originality:**
The work is original in that it is, to the best of my knowledge, the first that discusses non-asymptotic convergence and provides convergence bounds for MCSA methods. The paper casts MCSA methods as MCGD to leverage recently established results on non-asymptotic convergence for MCGD. Hence the theoretical contribution lies in deriving bound for the gradient variance and mixing rate of the individual methods such that they can be used with the existing theoretical findings. I believe this work is interesting to the inference community and might help to inform the further development of MSCA algorithms and their analysis.

**Clarity:**
Generally, due to the technical nature of the paper it was not always straightforward to me to follow the argument. I found it especially hard to follow the discussion about the convergence rates. The authors are mainly interested in the rates in terms of the gradient bound $G$, mixing rate $\rho$, and number of iterations $T$ and hence can express the convergence rates in a simplified form in O-notation. Even though the authors gave references to the exact theorems and corollaries which establish the convergence rates in previous work it was not straightforward for me to verify the exact correspondences, especially for the Mirror descent case (please see questions below).

**Significance**
Given the theoretical nature of the paper, I think the empirical evaluation sufficiently demonstrates the improvement of pMCSA over JSA and MSC. The first experiment seems to confirm the theoretical results but only considers two different sampling budgets. If computationally feasible, it would be nice to see an array of different sampling budgets, e.g. N=8,16,32,62,128, to further confirm the theoretical findings empirically.

---

> ### Author Response · Authors · 2022-08-01
> **Clarification on the Convergence Rates**
>
> Thank you for your important questions. We clarified them below.
>
> > Significance Given the theoretical nature of the paper, I think the empirical evaluation sufficiently demonstrates the improvement of pMCSA over JSA and MSC. The first experiment seems to confirm the theoretical results but only considers two different sampling budgets. If computationally feasible, it would be nice to see an array of different sampling budgets, e.g. N=8,16,32,62,128, to further confirm the theoretical findings empirically.
>
> Thank you for the suggestion. We have changed the plot to reflect a broader range of computational budgets in the **uploaded revision**.
>
> > I was specifically wondering about the $\log T$ term in the numerator of the rate for Mirror Descent which I was not able to match to the rate stated in the original paper.
>
> The stated convergence rate is not explicitly spelled out in the original paper. However, given that it is a direct consequence of Corollary 3.5 (Duchi et al., 2012), it was not clear whether it was worth including the full derivation in our paper. The stated convergence rate is obtained by setting $\epsilon = T^{-1/2}$, $\kappa_1 = 2 / \log \rho^{-1}$, $\kappa_2 = 1$. The $\kappa_1, \kappa_2$ terms are directly obtained from the definition of the Hellinger and total-variation mixing times (Definition 2.1; Duchi et al., 2012), while the arbitrary choice for $\epsilon$ is suggested by Duchi et al. (2012). Then, the term responsible for the convergence rate in Corollary 3.5 evolves as
> \begin{align}
> \frac{2 \alpha G^2}{\sqrt{T}} \left(\kappa_1 \log \frac{\kappa_2}{\epsilon} \right) = \frac{2 \alpha G^2}{\sqrt{T}} \left( \frac{2}{\log \rho^{-1}} \log \sqrt{T} \right) = \frac{2 \alpha G^2}{\log \rho^{-1}} \frac{\log T}{\sqrt{T}}
> \end{align}
> which, excluding the stepsize $\alpha$, is our stated rate.
> This is slightly worse than the rate stated in Eq. (3.2) by Duchi et al. (2012), but this is obtained through a stepsize aware of $G$, $\kappa1$, $\kappa2$, and the domain radius, which is less realistic.
>
> > I was also confused by the statement that the influence of mixing rate on the convergence rate decreases as $1/T^2$ even though the stated convergence rate does depend on the mixing rate at all. I might be missing something here and would be grateful for clarification.
>
> For Doan (2020ab), the convergence rate is "independent" of $T$ in the sense that the effect of the mixing time bound $\rho$ decreases at a faster rate. Therefore, the second and third row of Table 1 does not involve $\rho$.
>
> In more detail, the "effect" of the mixing rate by Doan (2020b) is contained in the positive integer $\mathcal{K}^*$. For the convex cases, for $k > \mathcal{K}^*$, the term with $\mathcal{K^*}$ decreases as $\mathcal{O}\left(1/k^2\right)$ where $k$ is the MCGD iteration number.  This is more explicitly stated by Doan (2020b) in Eq. (27), (15). Although $\mathcal{K}^*$ could be big, the stated rate is satisfied as soon as $k$ goes above it.

---

> > ### Comment · Reviewer_PECh · 2022-08-09
> > **Thanks for your response**
> >
> > Thank you for addressing my concerns. I increased my score to an *Accept*.

---

### Meta-Review · Area_Chair_jzVK · 2022-08-22

**Recommendation:** Accept
**Confidence:** Certain

**Metareview:**

All reviewers recommend accepting the paper. If the authors want to increase the impact of their work, a demonstration on a large-scale problem would help a lot.

**Award:**

No

---

### Decision · Program_Chairs · 2022-09-14

Accept